



# Sensitivity of modeled snow stability data to meteorological input uncertainty

Bettina Richter[1], Alec van Herwijnen[1], Mathias W. Rotach[2], and Jürg Schweizer[1]

[1]WSL Institute for Snow and Avalanche Research SLF, Davos, Switzerland
[2]Institute for Atmospheric and Cryospheric Sciences, University of Innsbruck, Innsbruck, Austria

**Correspondence:** Bettina Richter (richter@slf.ch)

**Abstract.** To perform spatial snow cover simulations for numerical avalanche forecasting, interpolation and downscaling of meteorological data are required, which introduce uncertainties. The repercussions of these uncertainties on modeled snow stability remain mostly unknown. We therefore assessed the contribution of meteorological input uncertainty on modeled snow stability by performing a global sensitivity analysis. We used the numerical snow cover model SNOWPACK to simulate two

snow instability metrics, i.e. the skier stability index and the critical crack length, for a field site equipped with an automatic weather station providing the necessary input for the model. Uncertainty ranges for meteorological forcing covered typical differences observed within a distance of 2 km and an elevation change of 200 m. Three different scenarios were investigated to better assess the influence of meteorological forcing on snow stability during a) the weak layer formation period, b) the slab formation period, and c) the weak layer and slab formation period. For each scenario, 14'000 simulations were performed,

by introducing quasi-random uncertainties to the meteorological input. Results showed that a weak layer formed in 99.7% of the simulations, indicating that the weak layer formation was very robust due to the prolonged dry period. For scenario a), modeled grain size of the weak layer was mainly sensitive to precipitation, while the shear strength of the weak layer was sensitive to most input variables, especially air temperature. Once the weak layer existed (case b), precipitation was the most prominent driver for snow stability. The sensitivity analysis highlighted that for all scenarios, the two stability metrics were

mostly sensitive precipitation. Precipitation determined the load of the slab, which in turn influenced weak layer properties. For case b) and c), the two stability metrics showed contradicting behaviours. With increasing precipitation, i.e. deep snowpacks, the skier stability index decreased (less stable). In contrast, the critical crack length increased with increasing precipitation. With regard to spatial simulations of snow stability, the high sensitivity on precipitation suggests that accurate precipitation patterns are necessary to obtain realistic snow stability patterns.

## 1  Introduction

Snow avalanches are a natural hazard, which can endanger roads, villages and human lives. A dry-snow slab avalanche starts with failure within a weak layer (Schweizer et al., 2003a). Such weak layers often form close to the snow surface. If subsequently, weak layers are covered by new snow, they can persist the entire season. Whether a failure in a weak layer is prone to propagate, depends on the complex interaction between slab layers and the weak layer (van Herwijnen and Jamieson, 2007).





The two key processes in avalanche release, failure initiation and crack propagation, can respectively be described with a stress-strength approach (expressed as stability index) and a fracture mechanical approach (considering the critical crack length as observed in a propagation saw test) (Reuter and Schweizer, 2018; Schweizer et al., 2016).

When assessing the avalanche danger, avalanche forecasters rely on snow instability data, combined with measured and forecasted meteorological data (McClung and Schaerer, 2006). Data on snow instability includes recent observations of avalanches,

or whumpfs and shooting cracks (Jamieson et al., 2009). Such signs of instability are very rare, especially on days with low avalanche activity (Reuter et al., 2015). Information on snow stratigraphy and so-called stability tests then becomes important. Unfortunately, these manual observations are relatively time-consuming point observations and sometimes dangerous to obtain so that the temporal and spatial resolution of snowpack data is limited. Detailed snow cover models, which simulate the full snowpack stratigraphy, can help fill this gap (e.g. Lafaysse et al., 2013; Morin et al., 2020) provided they include information

on snow instability (e.g. Schweizer et al., 2006; Lehning et al., 2004; Vernay et al., 2015).

The two most advanced snow cover models are Crocus (Brun et al., 1992; Vionnet et al., 2012) and SNOWPACK (Lehning et al., 2002; Wever et al., 2015). SNOWPACK can be used for one-dimensional simulations or for distributed snow cover modeling, when coupled with the three-dimensional model Alpine3D (Lehning et al., 2006). Crocus is part of the French model chain SAFRAN–SURFEX/ISBA-Crocus–MEPRA (S2M), which predicts the regional avalanche danger (Durand et al., 1999;

Lafaysse et al., 2013). The meteorological model SAFRAN provides the input for Crocus, which simulates the stratigraphy on virtual slopes for different elevations and aspects. MEPRA is an expert system, which derives the avalanche danger by combining various stability indices with a set of rules to evaluate the simulated snow stratigraphy in terms of stability classes (Giraud and Navarre, 1995). Recently, Vernay et al. (2015) drove S2M with an ensemble of atmospheric forcings to estimate the uncertainties from numerical weather prediction models. Meteorological input clearly influenced the forecasted avalanche

hazard and was assumed to be the main source of uncertainty. How these uncertainties influenced snow stability in more detail was not investigated.

The snow cover model SNOWPACK is forced with meteorological data from either automatic weather stations (Lehning et al., 1999) or numerical weather prediction models (Bellaire et al., 2011), and snow instability metrics can be derived from simulated stratigraphy (Lehning et al., 2004). The skier stability index $SK_{38}$ relates to failure initiation and compares the

shear strength of a weak layer with the shear stress acting on the weak layer due to the load of the overlaying slab and a skier (Föhn, 1987; Jamieson and Johnston, 1998; Monti et al., 2016). The critical crack length relates to crack propagation and was implemented into SNOWPACK by Gaume et al. (2017) and refined by Richter et al. (2019).

For numerical avalanche forecasting it is of particular importance how sensitive these stability criteria are to meteorological input uncertainty. Uncertainties due to spatial interpolation of meteorological data may arise when modeling distributed snow

stability. However, only a few studies have so far assessed the uncertainty of snow cover models. Côté et al. (2017) investigated the sensitivity of modeled snow height to three different weather models for five different automatic weather stations. They found that differences in forecast precipitation influenced modeled snow height. Bellaire et al. (2011) forced the SNOWPACK with output from a numerical weather prediction model. They showed that the weather prediction model generally overestimated precipitation events above 3 mm and therefore proposed different filtering methods for forecasted precipitation, which





influenced modeled snow depths. By applying a constant scaling factor for forecasted precipitation, SNOWPACK could repro-
duce measured snow depths and critical snow layers. Schlögl et al. (2016) systematically investigated the impact of different
model setups on the robustness of modeled snow water equivalent. They forced the distributed model Alpine3D with data from
automatic weather stations and showed that the coverage of weather stations can influence modeled snow water equivalent by
up to 20 %. Furthermore, they showed that decreasing model resolution from 25 m to 1000 m increased snow water equivalent
by up to 10 %. Lafaysse et al. (2017) developed a multi-physical ensemble system to estimate the uncertainty in modeled snow
height, density and albedo resulting from different physical parameterizations within a snow-cover model. Raleigh et al. (2015)
investigated how different error types, magnitudes and distributions of meteorological input parameters influenced simulated
snow water equivalent, ablation rates, snow disappearance and ablation. They employed a global sensitivity analysis based on
variance decomposition, which allowed to investigate the fractional contribution of different input parameters on the output
of non-linear models. Sauter and Obleitner (2015) performed a similar analysis to explore the influence of input uncertainty
on surface-energy balance components of snow cover models. Günther et al. (2019) investigated the sensitivity of snow water
equivalent at a field site in Austria to different sources of uncertainty, i.e. forcing errors, model structure and parameter choice.
They showed that forcing errors had the highest impact and parameter choice the lowest. However, no study so far addressed
the sensitivity of modeled snow instability estimates.

We therefore investigated how meteorological input uncertainty influenced modeled snow stability employing a global sen-
sitivity analysis. SNOWPACK was forced with meteorological input of an automatic weather station from a field site above
Davos, Switzerland and biases were introduced to the meteorological data. We performed simulations for the winter season
2016-2017, when one weak layer persisted for the entire season and affected snow stability in the region of Davos. We ana-
lyzed modeled snow instability metrics related to this weak layer in three steps: we independently investigated the influence
of meteorological input uncertainty during three periods of a) weak layer formation, b) slab formation, and c) weak layer and
slab formation.

The paper is organized as follows. Section 2 provides an overview of the study site and the simulations with SNOWPACK.
This is followed by the description of the uncertainties introduced to the model and the global sensitivity analysis. In Section 3,
we first shortly present the winter evolution. Then, the sensitivity of modeled slab and weak layer properties to uncertainties
in meteorological input is analysed for two different days: immediately after burial of the weak layer (Section 3.2.1) and for
a day with high avalanche activity (Section 3.2.2). Eventually, the evolution of snow stability was analysed with respect to
its sensitivity to input uncertainties during the three different periods (Section 3.3). Specific points are finally discussed in
Section 4.

## 2 Methods

### 2.1 Study site and data

We used data from the field site Weissfluhjoch (WFJ), located in the eastern Swiss Alps above Davos, at an elevation of
2536 m a.s.l. The WFJ site is equipped with an automatic weather station (AWS), which provides the necessary meteorological





input to the snow cover model. In addition, traditional snow profiles and stability tests were conducted weekly. Furthermore, we also calculated the Avalanche Activity Index (AAI) based on visual avalanche observation from the region of Davos (about 360 km$^2$), which were compiled by the avalanche warning service at the SLF. The AAI is the weighted sum of all observed avalanches, where weights are assigned according to avalanche size (Schweizer et al., 2003b). The winter season 2016-2017 was selected for this study, since the snowpack was marked by a prominent weak layer and pronounced avalanche activity on 9 March 2017. The weak layer formed between mid November 2016 and beginning of January at the surface of the shallow snowpack. For the analysis we will focus on the formation and evolution of this particular layer and its effect on snow stability for the period of high avalanche activity on 9 March 2017.

## 2.2 SNOWPACK

We performed simulations with the snow cover model SNOWPACK version v1473 (e.g. Lehning et al., 2002). SNOWPACK was driven with meteorological data from the AWS at WFJ, including precipitation (P), air temperature (TA), relative humidity (RH), wind velocity (VW), incoming shortwave (ISWR) and longwave (ILWR) radiation. For the reference run we used data from the quality controlled data set at WFJ (WSL Institute for Snow and Avalanche Research SLF, 2015). For the sensitivity analysis, we introduced uncertainties to the meteorological input. SNOWPACK calculated the absorbed shortwave radiation from modeled surface albedo, not from measured data. Furthermore, data on measured snow height and snow surface temperature was explicitly excluded in the configuration. The snow surface temperature was estimated from energy fluxes using Neumann boundary conditions at the snow-atmosphere interface (Bartelt and Lehning, 2002; Lehning et al., 2002). A constant geothermal heat flux of 0.06 W m$^{-2}$ was assumed at the bottom of the snowpack (Davies and Davies, 2010; Pollack et al., 1993). The time step for the simulation was 15 min and output was written every 24 h.

The sensitivity analysis focused on weak and slab properties, as well as modeled snow stability. In particular, the skier stability index $SK_{38}$ and the critical crack length $r_c$ were analyzed. We focused on the weak layer that formed between 16 November 2016 and 2 January 2017 (see red area in Figure 1). Since SNOWPACK produces considerably more layers than observed, all simulated snow layers that were deposited between these two dates and consisted of either depth hoar, surface hoar, facets and rounding facets were considered as weak layer, similar to Richter et al. (2019). Then weak layer properties were obtained by a thickness-weighted average $\bar{y}$ of the layer properties $y_i$:

$$\bar{y} = \frac{\sum y_i \, d_i}{\sum d_i},$$

where $d_i$ is the thickness of the simulated layer $i$. In analogy, slab properties were calculated from all layers above the weak layer, independent of grain type (see green area in Figure 1).

The $SK_{38}$ was calculated from layer properties of flat field simulations, which were extrapolated to a 38 $^\circ$ slope according to Jamieson and Johnston (1998)

$$SK_{38} = \frac{\tau_p}{\tau_s + \Delta\tau}, \tag{1}$$

with the shear strength $\tau_p$, the shear stress due to slab weight $\tau_s = \rho_{sl} g D_{sl} sin(38°) cos(38°)$, the average slab density $\rho_{sl}$, the slab thickness $D_{sl}$, the gravitational acceleration $g$, and the additional shear stress acting on the weak layer due to the





weight of a skier $\Delta\tau$. The additional shear stress is modeled as a line load (Föhn, 1987) and for a $38\,°$ slope, it simplifies to

120 $\Delta\tau = 155/D_{sl}\,\mathrm{m\,Pa}$ (Monti et al., 2016). Parameterizations for shear strength for different grain types were derived based on shear frame measurements (see Table 8 in Jamieson and Johnston, 2001) and implemented into SNOWPACK. For surface hoar, the shear strength was calculated according to Lehning et al. (2004). Details on shear strength parameterization in SNOWPACK were described by Richter et al. (2019).

The critical crack length was calculated from modeled layer properties using the improved parameterization suggested by

125 Richter et al. (2019):

$$r_c = \sqrt{F_{wl}}\sqrt{E'D_{sl}}\sqrt{\frac{2\tau_p}{\sigma_n}}, \tag{2}$$

with the plane strain elastic modulus of the slab $E' = \frac{E}{(1-\nu^2)}$, the Poisson's ratio of the slab $\nu = 0.2$, and the normal stress $\sigma_n = \rho_{sl}gD_{sl}$ acting on the weak layer due to the overlying slab. The elastic modulus of the slab, $E$, was related to the slab density by a power law fit to the data collected by Scapozza (2004):

$$E = 5.07 \times 10^9 \left(\frac{\rho_{sl}}{\rho_{ice}}\right)^{5.13}\,\mathrm{Pa}, \tag{3}$$

The correction factor $F_{wl}$ was introduced by Richter et al. (2019):

$$F_{wl} = 4.66 \times 10^{-9} \left(\frac{\rho_{wl}\,gs_{wl}}{\rho_{ice}\,gs_0}\right)^{-2.12}\,\mathrm{m\,Pa^{-1}}, \tag{4}$$

with the weak layer density $\rho_{wl}$, the weak layer grain size $gs_{wl}$, the density of ice $\rho_{ice} = 917\,\mathrm{kg\,m^{-3}}$ and the reference grain size $gs_0 = 0.00125\,\mathrm{m}$.

## 2.3 Forcing uncertainties

Uncertainties in the measured meteorological data (Table 1) were introduced based on the values suggested by Raleigh et al. (2015). Uncertainties can be seen as a systematic bias with a given range and distribution. The probability distributions of the biases were described by mean (normal: 1, lognormal: 20) and standard deviation (normal: 1, lognormal: 0.5) and then scaled within the given ranges. The bias $b$ was added to the forcing $F$ as $F' = F + b$ for an additive bias and $F' = F(1+b)$

for a multiplicative bias. Raleigh et al. (2015) proposed a multiplicative bias for precipitation (P) and an additive bias for air temperature (TA), relative humidity (RH), wind velocity (VW) and incoming longwave radiation (ILWR). For incoming shortwave radiation (ISWR) we chose a multiplicative bias using a range of $40\,\%$ according to the findings of Helbig and Löwe (2012). Biases resulting in non-physical forcing values were filtered to a physical range (e.g. RH was filtered within a range of [0,100] %).

In the reference run we used the data from the AWS at WFJ to drive the simulations. Then biases were introduced to the input data using three different scenarios. First, we introduced biases during weak layer formation up to the date when the weak layer was covered with new snow. The subsequent slab formation process occurred under the same conditions as in the reference



**Table 1.** Input uncertainties introduced as bias b to meteorological forcing.

| Forcing F | Distribution | Range | Unit | Perturbed forcing F' |
|---|---|---|---|---|
| P | Lognormal | [-75,+300] | % | F' = F(1 + b) |
| TA | Normal | [-3.0,+3.0] | °C | F' = F + b |
| RH | Normal | [-25,+25] | % | F' = F + b |
| VW | Normal | [-3.0,+3.0] | m s$^{-1}$ | F' = F + b |
| ISWR | Normal | [-40,+40] | % | F' = F(1 + b) |
| ILWR | Normal | [-25,+25] | W m$^{-2}$ | F' = F + b |

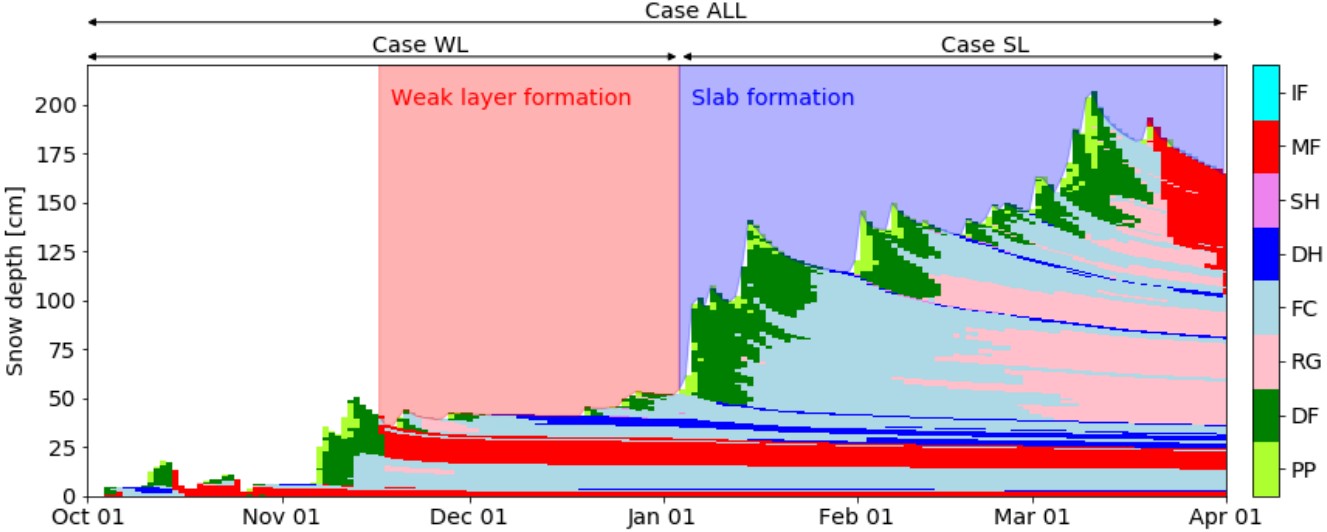

**Figure 1.** Reference run simulated with SNOWPACK for winter season 2016-2017 at the WFJ field site above Davos, Switzerland. Shown is the temporal evolution of simulated snow stratigraphy. Colors indicate grain type, i.e. precipitation particles (PP), decomposing and fragmented precipitation particles (DF), rounded grains (RG), faceted crystals (FC), depth hoar (DH), surface hoar (SH), melt forms (MF) and ice formations (IF). Red colored period refers to weak layer formation, blue colored period to slab formation. Arrows indicate different scenarios for which uncertainties were introduced into meteorological model input.

run. We refer to this first scenario as case WL (Figure 1). Second, meteorological conditions during the period of weak layer formation were identical to those of the reference run, while uncertainties were introduced during the period of slab formation
(case SL). Third, we introduced uncertainties to meteorological forcing during the entire simulation period (case ALL). For each scenario, 14,000 simulations were performed.





## 2.4 Global sensitivity analysis

Several studies have shown the advantages of considering co-existing sources of uncertainty by using a global sensitivity analysis rather than varying one input factor at a time while keeping all others fixed (Raleigh et al., 2015; Sauter and Obleitner, 2015). Following their approach, we employed a global sensitivity analysis to analyze the influence of input uncertainty to modeled snow instability. Sobol' (1990) suggested a robust method for nonlinear models based on variance decomposition. The total-order sensitivity index ($S_{Ti}$) was calculated as:

$$S_{Ti} = \frac{E\left[V(\mathbf{Y}|\mathbf{X}_{\sim i})\right]}{V(\mathbf{Y})} = 1 - \frac{V\left[E(\mathbf{Y}|\mathbf{X}_{\sim i})\right]}{V(\mathbf{Y})}, \tag{5}$$

where $E$ is the expectation operator, $V$ is the variance operator, $\mathbf{Y}$ is the model output and $\mathbf{X}_{\sim i}$ are all input parameters except $\mathbf{X}_i$. In our study, $S_{Ti}$ describes the variance in output variables, i.e. snow properties, $\mathbf{Y}$ due to uncertainties, introduced to a specific meteorological input $\mathbf{X}_i$, while including interactions with other forcing errors. Values for $S_{Ti}$ range from 0 to 1. For a perfect additive model, the sum of $S_{Ti}$ is equal to 1, otherwise it is greater than 1.

To efficiently compute $S_{Ti}$, a quasi-random set of input uncertainties was generated (Saltelli and Annoni, 2010; Saltelli et al., 2010). For this, two independent matrices of input uncertainties $\mathbf{A}$ and $\mathbf{B}$ were defined with the elements $a_{ji}$ and $b_{ji}$. The subscript $i$ ranges from one to the number of parameters $k$, in our case $k = 6$ is the number of forcings F (see Table 1). The subscript $j$ ranges from one to the number of samples N. The calculation of $S_{Ti}$ required the perturbation of parameters, so a third matrix $\mathbf{A}_B^{(i)}$ was introduced, where all columns were taken from $\mathbf{A}$, except for the $i$th column, which was taken from $\mathbf{B}$, resulting in a $kN \times k$ matrix. From Eq. (5) , $S_{Ti}$ can be computed as:

$$S_{Ti} = \frac{\frac{1}{2N}\sum\limits_{j=1}^{N}\left[f(\mathbf{A})_j - f\left(\mathbf{A}_B^{(i)}\right)_j\right]^2}{V(\mathbf{Y})}, \tag{6}$$

where $f(\mathbf{A})$ is the output variable evaluated on the $\mathbf{A}$ matrix and $f(\mathbf{A}_B^{(i)})$ is the output variable evaluated on the $\mathbf{A}_B^{(i)}$ matrix. For the calculation of $S_{Ti}$, we generated $N(2k + 2)$ samples, with $N = 1000$ base samples, resulting in 14,000 simulations.

## 3 Results

### 3.1 Winter evolution

The winter started with a snow storm accumulating around 50 cm of snow at the beginning of November 2016 (Figure 1). A melt-freeze crust subsequently formed at the snow surface due to high air temperatures between 16 November 2016 and 19 November 2016 at around 25 cm from the ground (Figure 1). This crust was also reported in manually observed snow profiles (not shown). Until 2 January 2017, 20 cm of snow accumulated above the crust (Figure 1). As the weather was mostly clear, the shallow snowpack was subject to strong temperature gradients during that period. The snow above the crust transformed into a weak layer of faceted crystals and depth hoar, which persisted throughout the entire season 2016-2017. This layer was visible in the simulated stratigraphy between 25 cm and 35 cm. After 2 January 2017, another 50 cm of snow accumulated, such that




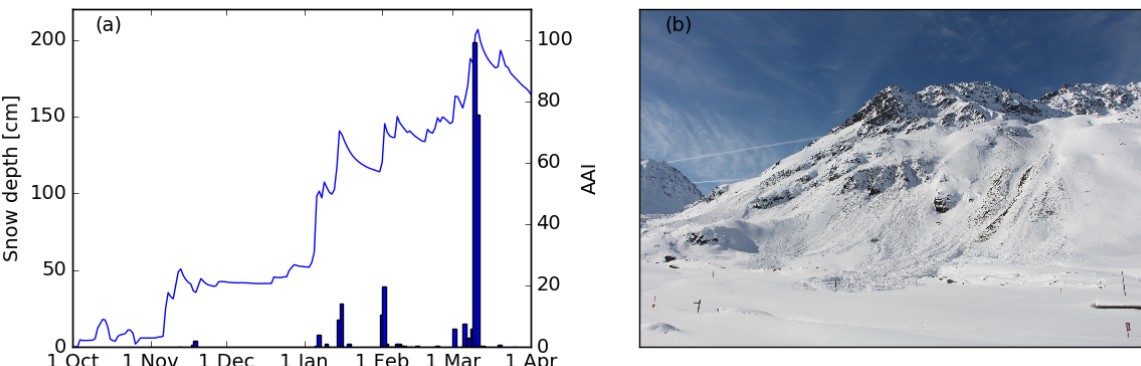

**Figure 2.** (a) Evolution of snow depth (full line) for winter season 2016-2017 at the WFJ field site above Davos, Switzerland and avalanche activity index (AAI) observed in the region of Davos (blue bars). (b) Avalanches that released during the cycle of 9-10 March 2017, in the valley of Dischma, Davos (picture taken on 15 March 2017). Often the ground or rocks are visible on the bed surface. This was a typical phenomenon for the winter season 2016-2017 due to the old snow problem.

the snow height increased from 50 cm to 100 cm within two days. Several small snow storms followed until a maximum snow height of about 200 cm was reached on 10 March 2017. Although the snow height only increased around 50 cm between 4 March 2017 and 10 March 2017, the peak of avalanche activity was observed by the end of this precipitation period during 9 March (Figure 2a). Many very large avalanches released during this period. Many avalanches in the region of Davos entrained

the whole snowpack so that the ground and rocks were visible on the bed surface (Figure 2b). As there were no fracture line profiles recorded, we cannot know in which weak layer the primary failure occurred. Since the weak layer that had formed in December 2016 was the most prominent persistent weak layer within the snowpack (Figure 1) this weak layer may have been the critical weakness. However, it is also possible that the primary failure occurred between new snow and old snow surface and then stepped down and entrained much of the old snowpack.

**3.2 Properties of weak layer and slab**

To quantify the influence of input uncertainty on slab and weak layer properties for the three cases, we focused on two specific points in time: 2 January 2017 when we investigated weak layer properties before burial and 9 March 2017 when avalanche activity peaked in the region of Davos (Heck et al., 2019).

**3.2.1 2 January 2017**

Up to 2 January 2017 the weak layers for case SL were identical to the reference run and for case WL and case ALL we used the same bias distributions. We therefore only present results for case WL and the reference run.

In the reference run, 95 % of the layers that had formed between 16 November and 2 January consisted of faceted grains with a mean grain size of 1.3 mm and a density of 188 kg m$^{-3}$ on 2 January 2017 (Triangles in Figure 3). For case WL, 36 %

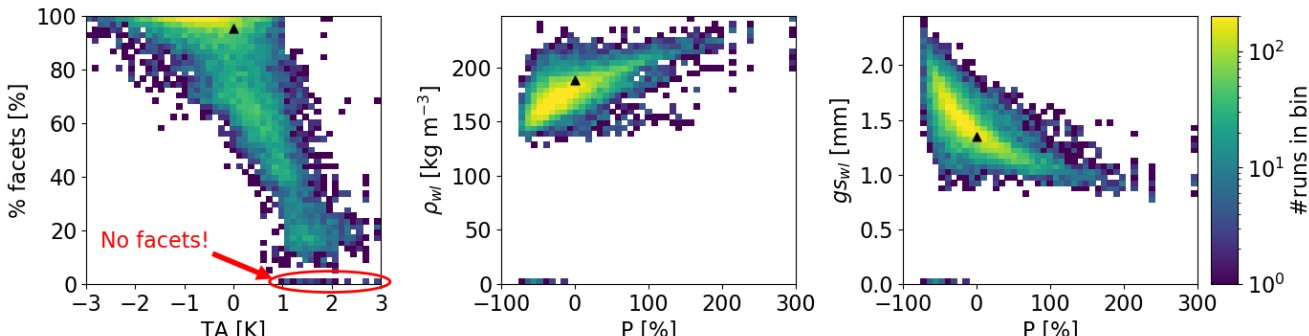

**Figure 3.** (a) Proportion of faceted layers within the weak layer with uncertainty in air temperature (TA) on 2 January 2017. (b) Density ($\rho_{wl}$) and (c) grain size ($gs_{wl}$) of faceted layers with uncertainty in precipitation (P). Colors indicate the binned number of simulations. Triangles indicate the reference run. Red ellipse indicates simulations, in which no weak layer formed.

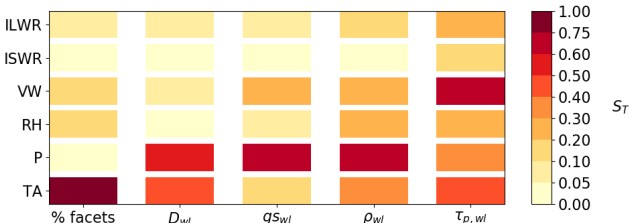

**Figure 4.** Total sensitivity index of weak layer variables on meteorological input uncertainty on 2 January 2017. Weak layer variables are the proportion of faceted layers (% facets), weak layer thickness ($D_{wl}$), weak layer grain size ($gs_{wl}$), weak layer density ($\rho_{wl}$) and weak layer shear strength ($\tau_{p,wl}$).

of the 14,000 simulations also predicted that at least 95 % of the layers that had formed between 16 November and 2 January

consisted of faceted grains (Figure 3a). In only 0.3 % of the simulations the weak layer did not form at all, i.e. there were no layers of faceted crystals. These simulations were characterized by a positive air temperature bias (red ellipse in Figure 3a). Warmer air temperature yielded less faceted layers within the weak layer and above a bias of $+1°$ C the percentage of faceted crystals occasionally reached 0 %. Overall, the percentage of faceted layers was highly sensitive to air temperature, while the thickness of the weak layer was sensitive to both TA and P (Figure 4). Grain size and density of the weak layer

were most sensitive to precipitation. Increasing P led to denser weak layers and smaller grains (Figure 3b,c). In fact, in 76 % of the simulations, the density of the weak layer was lower and in 67 % of the simulations the grain size was larger than in the reference run. Both properties, soft snow (low density) and larger grains are often associated with unstable weak layers (van Herwijnen and Jamieson, 2007). Finally, weak layer shear strength on 2 January 2017 was most sensitive to TA, P and VW ($S_T > 0.3$).




**Figure 5.** Modeled (a,d,g) weak layer density, (b,e,h) weak layer grain size and (c,f,i) load of the slab with uncertainty in precipitation (P) on 9 March 2017 for (a,b,c) case WL, (d,e,f) case SL and (g,h,i) case ALL. Colors indicate the binned number of simulations. Triangles indicate the reference run.


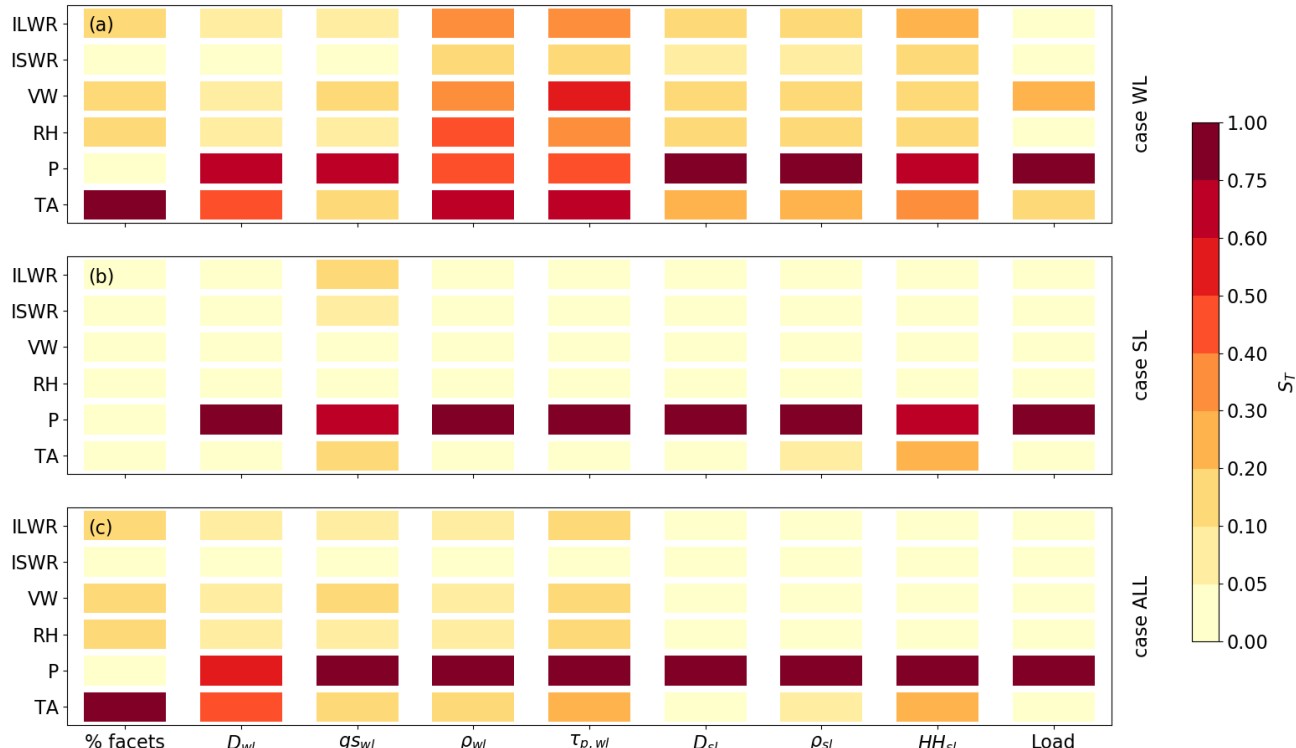

**Figure 6.** Total sensitivity index of different weak layer and slab variables on meteorological input uncertainty on 9 March 2017 for (a) case WL, (b) case SL and (c) case ALL. Variables are the proportion of facets (% facets), weak layer thickness ($D_{wl}$), weak layer grain size ($gs_{wl}$), weak layer density ($\rho_{wl}$), weak layer shear strength ($\tau_{p,wl}$), slab thickness ($D_{sl}$), slab density ($\rho_{sl}$), hand hardness index of the slab ($HH_{sl}$) and load due to the slab weight (Load).

### 3.2.2 9 March 2017

On 9 March 2017, mean weak layer density ($325\,\mathrm{kg\,m^{-3}}$) and mean grain size ($1.6\,\mathrm{mm}$) in the reference run had clearly increased compared to 2 January 2017. On top of the weak layer, the reference run simulated a 165 cm thick slab with a mean density of $256\,\mathrm{kg\,m^{-3}}$ corresponding to a load of $4.15\,\mathrm{kPa}$ (triangles in Figure 5).

In all three cases, around 66 % of the simulations predicted a weak layer with a lower mean density than in the reference run. The range was smallest for case WL, with $\rho_{wl}$ ranging from $295\,\mathrm{kg\,m^{-3}}$ to $370\,\mathrm{kg\,m^{-3}}$ and highest for case ALL, with $\rho_{wl}$ ranging from $240\,\mathrm{kg\,m^{-3}}$ to $401\,\mathrm{kg\,m^{-3}}$ (Figure 5a,d,g). This means that the weak layer density on 9 March 2017 was more influenced by the slab than the original density prior to burial. In contrast, the grain size of the weak layer rather depended on the original grain size. Hence, the dispersion for case ALL was similar to case WL, with $gs_{wl}$ ranging from $1.0\,\mathrm{mm}$ to $3.0\,\mathrm{mm}$, whereas $gs_{wl}$ predicted by case SL was similar to the reference run, ranging from $1.6\,\mathrm{mm}$ to $2.0\,\mathrm{mm}$ (Figure 5b,e,h). In all




three cases, around 70 % of the simulations predicted grain sizes larger than the reference run. As expected, the range in slab properties for case WL was minimal on 9 March 2017, e.g. the load of the slab ranged from 4.13 kPa to 4.18 kPa. In contrast, the load for case SL and case ALL varied by a factor of 16, ranging from 1.03 kPa to 16.7 kPa (Figure 5c,f,i). In all three cases, around one third of the simulations predicted a higher slab load. Other slab properties, e.g. slab density, did not vary much for case WL, whereas they greatly varied for case SL and case ALL. To sum up, different slab properties strongly influenced the evolution of the weak layers, whereas different weak layers, as expected, did not influence the evolution of the slab.

The total sensitivity indices for case WL on 9 March 2017 were similar to those on 2 January 2017 for $D_{wl}$ and $gs_{wl}$. For density and shear strength, all input parameters except ISWR increased to $S_T > 0.3$ (Figure 6a). In contrast, for case SL, weak layer and slab properties were dominantly sensitive to P (Figure 6b). Increasing P increased the load on the weak layer and yielded smaller grains and higher weak layer density (Figure 5d,e). For case ALL, uncertainties in P dominated weak and slab properties. Similarly, density and shear strength of the weak layer on 9 March 2017 were mostly sensitive to P, suggesting that the density evolution of the weak layer was determined by the load rather than the original density after burial.

### 3.3 Evolution of snow stability

After burial of the weak layer, snow stability of the reference run, i.e. $SK_{38}$ and $r_c$, initially increased with time (black lines in Figure 7). During periods with precipitation (increases in snow height in Figure 1), both indices decreased, whereas during periods without precipitation, both indices increased. However, this increase was very weak for $SK_{38}$. On 30 January 2017, $SK_{38}$ reached a maximum value of 1.24. After that, decreases in $SK_{38}$ during periods with precipitation events were stronger than increases $SK_{38}$ during periods without precipitation. Therefore, an overall decrease was observed for $SK_{38}$ after 30 January 2017, such that $SK_{38}$ reached a minimum value of 0.81 during the period of high avalanche activity (10 March 2017). In contrast, $r_c$ increased more prominently during periods without precipitation, such that $r_c$ reached a minimum value of 15 cm right after burial and a maximum value of 124 cm by the end of March. During periods with precipitation, $r_c$ decreased, e.g. $r_c$ decreased just before the period 9 March 2017, such that lower values for $r_c$ during the peak of avalanche activity were modeled (indicated by grey vertical bars in Figure 7). Therefore, days with high avalanche activity coincided with days with small values for $r_c$ and small values for $SK_{38}$ (Figure 8).

Similar to the reference run, $SK_{38}$ and $r_c$ initially increased for all three cases. After 30 January, an overall decrease in $SK_{38}$ was observed, while the increase in $r_c$ was more pronounced towards the end of the simulation period. During periods with precipitation, decreases in snow stability were observed (Figure 7). The range of $SK_{38}$ was larger in case SL compared to case WL, suggesting that the load due to slab weight had a stronger influence on $SK_{38}$ than the shear strength of the weak layer. On 9 March 2017 for instance, $SK_{38}$ ranged from 0.79 to 1.87 for case WL and 0.33 to 1.90 for case SL (Figure 9a,c). This suggests, that different slabs influenced $SK_{38}$ more than different weak layers, i.e. the slab was more important. Case ALL showed the largest range from 0.32 to 3.05 (Figure 9e). Around one third of the simulations for all cases predicted a $SK_{38}$ smaller than that for the reference run with a value of 0.86 (case WL: 44 %, case SL: 36 % and case ALL: 33 %). The spread of $r_c$ was similar in case WL and case SL, ranging from around 30 cm to 100 cm on 9 March 2017 (Figure 9b,d). For case ALL, the spread of $r_c$ was larger, ranging from 18 cm to 146 cm on 9 March 2017 (Figure 9f). Around two thirds of the



**Figure 7.** Temporal evolution (January to March 2017) of input uncertainties to modeled skier stability index $SK_{38}$ for (a) case WL, (c) case SL, (e) case ALL and critical crack length $r_c$ for (b) case WL, (d) case SL, (f) case ALL. Colors indicate the binned number of simulations. Black lines show the reference run and grey vertical bars highlight the period of high avalanche activity.

simulations in all three cases predicted a value of $r_c$ smaller than in the reference run with a value of 58 cm (case WL: 71 %,

case SL: 65 % and case ALL: 69 %). However, only 30 % of the simulations of case WL predicted both, a smaller $SK_{38}$ and a smaller $r_c$ value on 9 March 2017. For case SL and case ALL, only 6 % and 7 % of the simulations, respectively, predicted


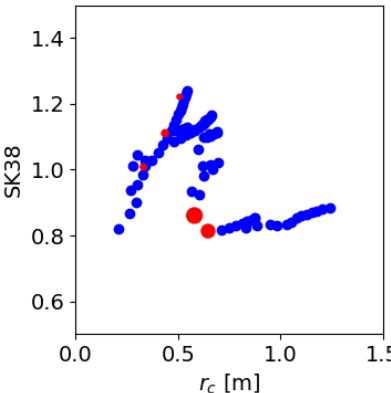

**Figure 8.** $SK_{38}$ with $r_c$ of the reference run for all days from 4 January 2017 to 31 March 2017. Red circles are days with AAI > 10 and the size of the circles correspond to the value of AAI.

lower values for both stability indices. This means that if a simulation yields a smaller $SK_{38}$, $r_c$ was mostly larger. Stability indices therefore did not respond to the biases in a similar manner.

While $r_c$ was mostly sensitive to precipitation, for case WL, $SK_{38}$ was highly sensitive to TA (Fig. 10a,b). In contrast,
for case SL and case ALL, the total-order sensitivity clearly highlighted precipitation as the most dominant input parameter for stability indices (Figure 10c-f). Although during precipitation events, $r_c$ temporarily decreased (Figure 7) the load by the slab affected the consolidation of the weak layer as well as the slab layers. A higher load induced higher weak layer strength and a stiffer slab so that $r_c$ increased. On 9 March 2017, for all cases, increasing precipitation yielded larger critical crack lengths (Figure 9). This strongest increase for $r_c$ with P was observed for case ALL (Figure 9d). Whereas $SK_{38}$ increased
with increasing TA for case WL (Figure 9a), it clearly decreased with increasing P for case SL and case ALL (Figure 9c,e). The decrease is a consequence of the more prominent increase in slab load than in shear strength. In fact, the shear strength increased with increasing precipitation by a factor of two while slab load increased with increasing precipitation by a factor of six (not shown).

## 4    Discussion

We examined the sensitivity of modeled snow stability to meteorological input uncertainty using a global sensitivity analysis approach suggested by Sobol' (1990). To do so, we introduced biases to six meteorological inputs: air temperature, relative humidity, precipitation, wind velocity, incoming short- and long wave radiation, which are all required as input variables by the snow cover model SNOWPACK (Lehning et al., 2002). Among these input parameters, precipitation had the most prominent influence on modeled snow stability. Precipitation influences weak and slab properties. Although a positive bias in

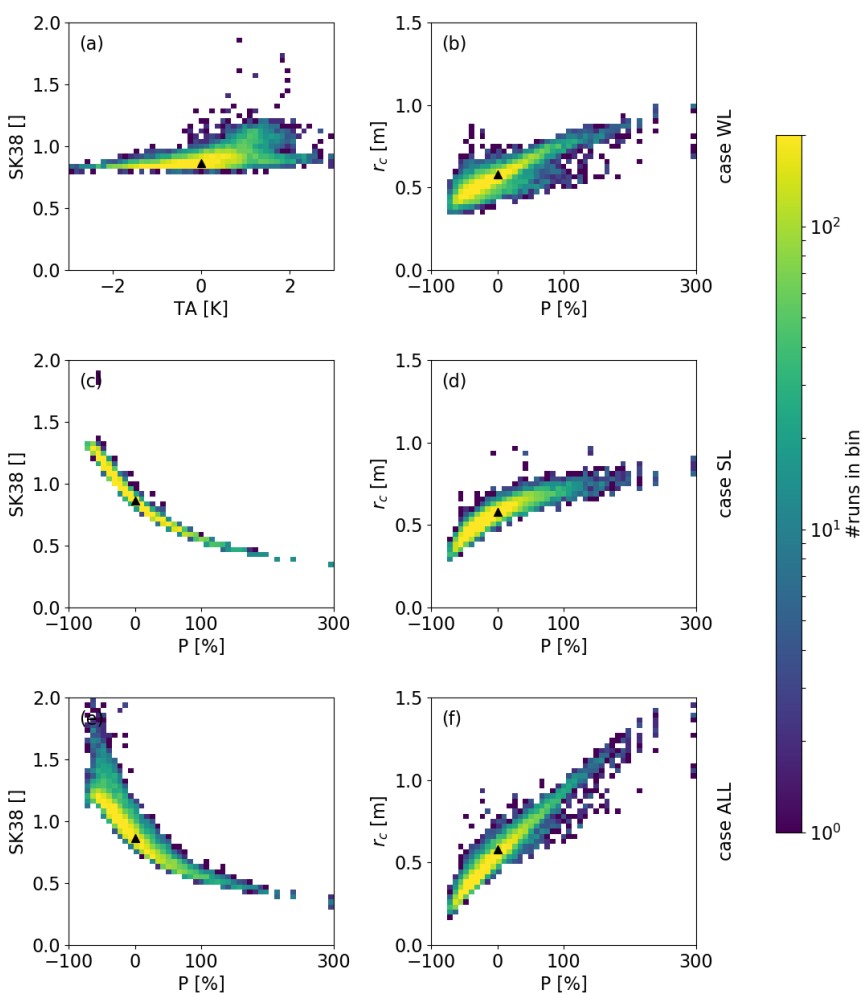

**Figure 9.** Modeled (a,c,e) skier stability index ($SK_{38}$) and (b,d,f) critical crack length with uncertainty in most sensitive input parameter, i.e. air temperature (TA) and precipitation (P) on 9 March 2017 for (a,b) case WL, (c,d) case SL and (e,f) case ALL. Colors indicate the number of simulations in each of the 50×50 bins. Triangles indicate the reference run.

air temperature reduced the percentage of faceted crystals within the weak layer, in most simulations a weak layer had formed, which may indicate a widespread avalanche problem in the region.

We used biases instead of random uncertainties, as Raleigh et al. (2015) investigated different sources of errors and showed that biases had more influence on model output. For the parameter biases, we used the ranges suggested by Raleigh et al. (2015), who provided a comprehensive overview of typical variations in these parameters in complex topography. The only exception



**Figure 10.** Evolution of the total sensitivity index $S_T$ for the model output $SK_{38}$ and $r_c$ for case WL, case SL and case ALL (from top to bottom). Grey vertical bars highlight period of high avalanche activity.

was for ISWR, for which we chose a multiplicative bias rather than a cumulative bias, since we expected bias in ISWR to depend on solar angle. As the radiation balance in snow covered complex topography can lead to large variations in incoming shortwave radiation (Helbig and Löwe, 2012), we used a range of $\pm 40\%$. While the snow cover model SNOWPACK has traditionally been forced with measured data from automatic weather stations (Lehning et al., 1999; Monti et al., 2015; Wever et al., 2015), it is increasingly used for spatially distributed model applications either by interpolating measured meteorological




data or using output from numerical weather prediction models (Bellaire et al., 2011; Bellaire and Jamieson, 2012; Schlögl et al., 2016). As such, the introduced biases can be seen as potential errors due to the interpolation schemes, or biases in the NWP output. For instance, for air temperature, the variation of $\pm 3\,°C$ (Table 1) corresponds roughly to typical errors between NWP output and TA measurements (Bellaire et al., 2017).

In complex terrain, wind induced processes strongly influence snow distribution (Mott and Lehning, 2010). The bias intro-
duced for P agree with the high variations in snow depths, measured at very small scale (Bühler et al., 2015). P had the most significant impact on modeled sensitivity, which may partly be due to the high magnitude of bias (Raleigh et al., 2015). These results have implications for spatial snow cover modeling, which is increasingly applied in avalanche forecasting (Bellaire et al., 2017, 2011; Morin et al., 2020; Lafaysse et al., 2017; Vernay et al., 2015). Indeed, our results suggest that if we want to obtain realistic spatial patterns, we need to adequately model snow distribution in mountainous regions. This is not an easy
task, as snow distribution is very complex (Grünewald et al., 2010; Helbig and van Herwijnen, 2017; Kirchner et al., 2014; Reuter et al., 2016). Since the mountain snow cover is largly shaped by snow transport by wind, adequate modeling can only be achieved through computationally expensive snow drift modeling (Gerber et al., 2018; Mott and Lehning, 2010; Vionnet et al., 2014). While from an operational point of view, high resolution modeling (resolution of several meters) on large domains is presently out of reach, alternative approaches were suggested (e.g. Helbig et al., 2017; Vögeli et al., 2016; Winstral et al.,
300   2002).

Previous snow sensitivity studies typically focused on snow depth or snow water equivalent by introducing model uncertain-
ties during the entire season (Lapo et al., 2015; Raleigh et al., 2015; Sauter and Obleitner, 2015). For most applications, such as snow hydrology or glacier mass balance, these target variables and time scales are sufficient. However, for snow instability assessment and avalanche formation the relevant time scales are shorter (weeks) and snow stratigraphy is a key variable that
has to be accounted for (Schweizer et al., 2003a). Indeed, a necessary pre-requisite for dry-snow slab avalanche release is a weak layer within the snow cover below a cohesive slab.

Simulations were performed for the field site Weissfluhjoch above Davos, Switzerland, for the winter season 2016-2017. This winter was characterized by a thick persistent weak layer that developed early in the season (December) and likely contributed to a widespread avalanche cycle in the area in March (Figure 2). We thus investigated the formation and subsequent
burial of this weak layer consisting of faceted grains and depth hoar near the base of the snow cover, often called persistent weak layer (Jamieson and Johnston, 1992; Schweizer et al., 2003a). Such early season weak layers are often widespread and associated with poor stability for most of the season. Grain size and hardness are important parameters to identify persistent weak layers and evaluate snow stability (e.g. Schweizer and Jamieson, 2007; van Herwijnen and Jamieson, 2007). Results from our sensitivity analysis thus showed that the formation of the weak layer was mostly influenced by precipitation and air
temperature early in the season (Figure 4). This comes as no surprise since both the parameters directly affect the temperature gradient across the snowpack, which is the most important driver for the formation of facets and depth hoar (Birkeland, 1998; Miller and Adams, 2009; Staron et al., 2012). Our results also show that the formation of persistent weak layers is rather robust. Indeed, in only 0.3 % of the simulations no weak layer developed, suggesting that even within the range of meteorological input we used, if a prolonged dry weather period occurs after the first snowfall, such weak layers will generally form. Only warm




weather can prevent the formation of a weak layer during a prolonged dry weather period, which is generally found at lower elevations. Our results suggest that spatial snow cover modeling can be used to predict the elevation range for weak layers. However, we only looked at one type of weak layer. The formation and subsequent burial of surface hoar might be more sensitive to other meteorological parameters, such as wind speed (Stössel et al., 2010).

     We focused on two metrics of snow instability, namely $SK_{38}$ (Eq. (1)) and $r_c$ (Eq. (2)). These metrics relate to both failure

initiation ($SK_{38}$) and crack propagation ($r_c$), two fundamental processes required for avalanche release (Reuter and Schweizer, 2018; van Herwijnen and Jamieson, 2007). Given our current understanding of snow stability, critical weak layers require both a low failure initiation propensity and a low crack propagation propensity (Reuter and Schweizer, 2018). While both these indices have been validated (Schweizer et al., 2006; Richter et al., 2019), thus far no threshold values exist that separate stable from unstable snow conditions adapted for use in SNOWPACK. As such, we compared these stability indices to the reference

run to determine if the introduced biases resulted in a more stable or a less stable snowpack, with a particular focus on 9 March 2017 when avalanche activity in the regions of Davos peaked (Fig 2).

     To better assess the role of slab and weak layer properties with respect to snow instability, we used three scenarios where we varied meteorological input only during the weak layer formation period, only during the slab formation period and during the entire period. These three scenarios clearly highlighted that weak layer and slab formation are sensitive to different

meteorological parameters and can influence snow instability in very different ways (Figures 6 and 10). With higher precipitation during the slab formation period $r_c$ generally increased, whereas $SK_{38}$ decreased. More precipitation resulted in thicker slabs which typically have a higher density, hardness and stiffness (van Herwijnen and Jamieson, 2007; van Herwijnen et al., 2016). Furthermore, due to higher slab load, the weak layer shear strength increased, resulting in higher values for $r_c$. This is in line with other parametric studies on snow instability showing that slab properties substantially affect snow instability

(Gaume et al., 2017; Reuter and Schweizer, 2018; Schweizer and Reuter, 2015). Furthermore, our results suggest that even if a persistent weak layer forms at the start of the season, the remainder of the winter season can still have a profound effect on the overall evolution of snow instability.

     In contrast, the decrease in $SK_{38}$ is a consequence of the more prominent increase in slab load than in shear strength. In fact, the shear strength increased with increasing precipitation by a factor of two, while slab load increased with increasing

precipitation by a factor of six (not shown). With increasing slab thickness the skier stress on the weak layer decreases and skier triggering becomes unlikely. $SK_{38}$ can no longer be used to assess skier triggering (Schweizer et al., 2016). Instead, other stability indices should be considered, e.g. the natural stability index. However, the denominator in Eq. (1) is dominated by the shear stress due to the load of the slab for thicker slabs. Hence $SK_{38}$ approaches the natural stability index for slab thicknesses above approximately one meter. During the precipitation event of 9 March 2017, these strength-over-stress approaches reach

a small value, meaning that a failure is easy to initiate. Even towards the end of March 2017, $SK_{38}$ and the natural stability index (not shown) remain very low, which is rather counter-intuitive regarding failure initiation.

     In the context of climate change, Castebrunet et al. (2014) suggested a decrease in avalanche activity for the Alps and an increase in wet-snow avalanche activity during winter at high elevations. Martin et al. (2001) assumed that avalanche hazard (number of days with moderate or high avalanche hazard) decreased with increasing TA. Our study also allows assessing the



effect of increasing temperature on snow instability. With increasing TA during the formation of the weak layer, the weak layer will get stronger, meaning higher density and smaller grain size. This results in an overall more stable snowpack. However, in our case study only 0.3 % of the simulations, no weak layer formed at all. We therefore expect, that instabilities due to persistent weak layers will continue to challenge avalanche forecasting. This is in particular of interest, since about 70 % of 186 skier-triggered avalanches were released in weak layers of persistent grain types, i.e. surface hoar, faceted crystals, and depth hoar

(Schweizer and Jamieson, 2001). The primary driver after weak layer formation was precipitation, with partly opposing effects on our snow instability metrics.

## 5   Conclusions

We investigated the sensitivity of two modeled snow instability metrics on meteorological input uncertainty employing a global sensitivity analysis. We evaluated three scenarios, in which uncertainties were introduced during the weak layer formation

period, the slab formation period and the whole winter season. This approach allowed to independently investigate the effects on weak layer and slab properties, which both contribute to snow stability.

The process of weak layer formation was very robust as in most simulations persistent grain types formed. However, weak layer properties strongly depended on meteorological conditions during the formation period. While weak layer grain size was sensitive to precipitation, weak layer density and shear strength were also sensitive to other input parameters during the period

of weak layer formation, such as air temperature, relative humidity and wind velocity. The smaller the strength of the weak layer initially was, the weaker the snowpack stayed later on. Once a weak layer had formed, its properties were strongly sensitive to uncertainties in precipitation during the slab formation period. While the grain size of the weak layer was more determined by the initial grain size before burial, the weak layer density and accordingly, weak layer shear strength were mostly determined by the load of the slab. Moreover, slab properties were largely sensitive to precipitation during slab formation. Therefore,

precipitation was found to be the strongest driver of snow properties. These snow properties, however, influenced modeled snow stability in different ways. For example, a positive bias in precipitation, which can be found in wind-shaded areas with above-average accumulation, resulted in an overall lower skier stability index and higher critical crack length. Vice versa for areas with below-average snow depth, a higher skier stability index and a lower critical crack length was simulated.

As snow deposition in complex terrain substantially varies during storms and given the high sensitivity of stability to pre-

cipitation, numerical forecasting of snow stability in 3D terrain will require spatially highly resolved precipitation patterns.

*Data availability.*   Upon acceptance all relevant data will be made available on www.envidat.ch.

*Author contributions.*   BR processed and analyzed the simulations. AH and JS initiated this study. BR prepared the paper with contributions from all co-authors.





*Competing interests.* The authors declare that they have no conflict of interests.

*Acknowledgements.* Thanks to Mathias Bavay for helping with SNOWPACK issues, Thomas Kramer for IT support and Henning Löwe for discussions on programming style. Bettina Richter has been supported by a grant of the Swiss National Science Foundation (200021_169641).



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
