# Peer review of "Sensitivity of modeled snow stability data to meteorological input uncertainty"

_Natural Hazards and Earth System Sciences, 2019_

## Referee Comment (RC1) · Simon Horton (Referee) · 4 Apr 2020

**General comments**

This paper investigates how stability indices predicted by snowpack models are impacted by uncertainties in the weather inputs. Spatial snowpack simulations could be valuable for avalanche forecasting, however there are numerous challenges in producing accurate spatially distributed weather inputs for these models. This paper provides a strong quantitative analysis of what the implication of these uncertainties are when assessing snowpack stability. The sensitivity analysis uses weather data for a situation where a prominent weak layer formed in the snowpack and subsequently resulted in avalanche activity throughout the season. Although only a single scenario is in-

vestigated, the implications of various biases added into the data provides a robust analysis of how uncertainties in different weather inputs impacts weak layer formation, slab properties, and snowpack stability.

The paper provides a significant contribution by improving the interpreting stability indices and illustrating the need for improved prediction of snowfall patterns. The methods are valid and rigorous, and the manuscript is well structured, organized, and easy to follow. My comments are relatively minor and could improve the manuscript by clarifying a few details and expanding on some interesting results.

**Specific comments**

- An interesting result that could use more discussion is explaining why the uncertainties resulted in unequal proportions of properties relative to the reference run. For example, Fig. 3 shows the majority cases had weak layers with lower densities and larger grain sizes than the reference run, and Fig. 9 shows the majority of cases had smaller critical crack lengths than the reference run. While such trends are reported throughout the results, they are not explained in the Discussion. Do these results mean (a) the distribution of input uncertainties were biases towards these results, (b) there were interaction effects between different combinations of biases that favoured these results, (c) some type of non-linearities in the model, (d) something else? If related to the biases, which biases resulted in these trends and why?

- There could be a bit more clarity on how the biases were applied to the weather data, since the distribution of weather inputs has a substantial effect on the results. I interpreted the method as follows: for a given time series, a bias $b$ was randomly chosen for each variable and then that single value applied to the variable for the entire season. This could be stated more explicitly. If random biases

were selected for each variable you would expect a roughly equal proportions of different bias combinations (e.g. samples with P+/TA+, P+/TA-, P-/TA+, P-/TA-). Would such combinations reflect the distribution of conditions you would actually expect to find in nature? Is this method consistent with other sensitivity studies using weather data? I suspect the method of applying these biases resulted in the skewed proportions discussed in the previous comment.

- A limitation of the study is that it considers a single type of weak layer and snow-pack structure combinations (i.e. early season facets above a crust). The type of weak layer considered in this study is important and should be stated in more places (e.g. abstract and conclusions). While briefly discussed in lines 317-323, many of the results likely still generalize to more types of snowpack conditions (especially the slab properties). For surface hoar, a major sensitivity is the ex-posure time of the layer on the surface in between precipitation events. A light amount of snow could stop surface hoar growth in a much more dramatic way than facets. This again strengthens the argument that precipitation patterns (spa-tial, quantity, and timing!) are critical. While the details of surface hoar formation are outside the scope of this study, acknowledgement of this limitation and more discussion of what results likely transfer to other weak layers would be valuable.

- While the paper touches on most of the interesting results, there are a few minor results listed in the Technical comments that could also be discussed (e.g. why does wind speed impact shear strength?, why does weak layer grain size on 2 Jan not show sensitivity to temperature or radiation as might be expected for facets?)

- The discussion section could be reduced as there is substantial repetition from previous sections (e.g. lines 281-282 repeat the methods, lines 301-306 repeat introduction/motivation of study, lines 307-309 repeat methods, etc.). While this section is well written and examines interesting results, the repetition of why and

how the study was done is unnecessary.

- The conclusions could have greater a emphasis on the contributions of the study. Although well written, they primarily focus is summarizing the results.

- Overall the figures are clear, legible, and are effective at communicating the key results of the study.

**Technical comments**

- p1 l8: It would be helpful for the abstract to briefly explain the snowpack conditions for the case study (especially the fact the type of weak layer was early season facets above a crust).

- p1 l 17: add "(more stable)" following crack length sentence for consistent structure.

- p1 l15 "sensitive **to** precipitation"

- p2 l49-52: It would be helpful to explicitly explain how to interpret SK38 and rc in relation to initiation and propagation (e.g. "low values of SK38 indicate initiation more likely, low values of rc indicate propagation more likely")

- p2-3 l 53-74: I appreciate how this paragraph concludes by identifying the clear gap in literature that this study addresses, however most of the paragraph reads like a long list of studies and the link to your research question isn't always apparent. I think by rewording some sentences it could be clearer how these studies relate to your research question. Also, Andrew Slaughter's PhD thesis (Slaughter, 2010) performs a SOBOL sensitivity analysis for formation of several types of weak layers and is relevant to this study.

- p4 l1: Just a comment: the weekly snow profiles aren't directly used in your study, although I assume they were important for understanding the avalanche conditions that you describe.

- p4 l115: Thickness-weighted averaging may smooth out the properties of the most unstable layer(s) that may contain the critical properties for avalanche release. Could this averaging method somehow impact the biases favouring the formation of more unstable layers?

- p4 l116: "shear strength **of the weak layer**..."

- p4 l116: I understand you present the SK38 and rc derivations in general form, but would it make sense to use the bar notation for the variables that you substitute with thickness-weighted averages (such as slab and weak layer densities)?

- p5 l131-134: Please provide a written explanation of what this correction factor accounts for.

- p5 l 136: In the abstract you specify the uncertainty values are typical for extents of 2 km and elevation changes of 200 m. It would be worth including that somewhere in the text.

- p6 l150: Please specify here whether Case ALL has a unique set of biases or simply concatenates the two other cases.

- Sect. 2.4: This section could use some additional explanation. First, it would be helpful to move the written description of what $S_{Ti}$ means (line 160-161) before the mathematical definition in Eq. 5. On line 162 you describe a 'perfect additive model', but do not explain whether this is important or how that idea applies to this study. It's not clear what information is contained in the $A$ and $B$ matrices as you simply describe their dimensions rather than their content, and thus the

importance of $A_B$ is unclear. Without explanation I'm wondering if $B$ is a matrix full of biases $b$ you introduce in Table 1 (i.e. the same letter).

- Sect. 3.1: This section provides a very clear and helpful practical explanation of the case study.

- p8 l195: A more intuitive wording would be something like "We present weak layer and slab properties on 2 January with results from the reference run and case WL because..."

- p8 l197: In the methods your weak layer group consists of more than just facets (e.g depth hoar and surface hoar), does the "percent facets" variable actually mean percent of weak layers or literally percent facets and there was no depth hoar or surface hoar?

- Fig. 3 and 4: Would a more logical progression be showing Fig. 4 first to show which input uncertainties had the greatest effect then show Fig. 3 to show the direction of the effect? Seeing which weather input had the greatest impact on a given property would help explain why a specific scatter plot is being shown. Same logic applies to Fig. 5 and 6 and 9 and 10. Just a thought.

- p9 l204: Is weak layer thickness also calculated as an average of each individual layer, or was it the sum of all identified weak layers? The sum seems more meaningful.

- p9 l204-205: This result about the impact of precipitation is somewhat unique to how this weak layer is being identified (as all layers forming over a date range), and it is not necessarily intuitive to think about how precipitation during a formation period impacts weak layer formation. It would be helpful to reiterate what precipitation means for this specific case. Also, wouldn't you expect grain size to be more sensitive to air temperature (and perhaps the radiation variables) given the weak layers were faceted crystals?

- p9 l208: It would be interesting to discuss why weak layer shear strength was most sensitive to wind speed as well as the direction of the relationship (i.e. did increasing wind typically result in higher or lower shear strength?). This result is not necessarily the most intuitive and could be discussed more.

- p11 l211: It would be helpful to introduce this date the same way as 2 Jan by introducing the fact you now consider all three cases before you start reporting results.

- Fig. 5: The load-P subplots present obvious results and it's not clear there's added value in graphing these relationships.

- p12 l223: High slab load than what? The reference case?

- p12 l227: Does it make sense that $S_T$ would change between 2 Jan and 9 mar if Case WL uses the reference data from 2 Jan onwards?

- p12 l233-243: This paragraph is very well written and easy to follow!

- Fig. 8: Is it correct to follow the points as a time series starting from the bottom left? If so, including a line connecting the points (and perhaps even an arrow) could make it clearer this shows evolving stability properties rather than an independent scatter of data points.

- p13 l 254: Could you provide a similar summary for rc as done for SK38 in line 247 ("This suggests, that different slabs influenced SK38 more than different weak layers"). It appears from Fig. 9 rc was equally impacted by weak layer and slab properties.

- p14 l259-268: I found this paragraph slightly confusing to read. Perhaps some parts could be reworded or even some of the interpretation moved to the Discussion.

- p18 l320-324: This result agrees with Horton et al. (2015) who examine how variability in meteorological fields from NWP models across elevations resulted in reasonable predictions of surface hoar formation. Slaughter (2010) also analyzes sensitives of surface hoar and other weak layers to weather inputs.

- p18 l340-342: This is a very practical take away from this study that supports practical forecasting experience, and could be a valuable application of snowpack models.

- p18 l343: "than in **weak layer** shear strength"

- p18 l344-345: These results could be supported by citing field studies that describe the lag in weak layer shear strength increases after loading, such as Jamieson et al. (2007) who also give interesting implications on spatial variability of stability indexes due to variable precipitation.

- 18 lp351: How do you explain this counter intuitive result where SK38 remains low into spring? It would seem that since the load continues to increase that the weak layer strength must have remained low. Was this the case?

- p19 l360-361: How does this sentence about precipitation tie back to the theme of climate change?

**References**

- Horton, S., Schirmer, M., and Jamieson, B.: Meteorological, elevation, and slope effects on surface hoar formation, The Cryosphere, 9, 1523–1533, https://doi.org/10.5194/tc-9-1523-2015, 2015.

- Jamieson, B., Zeidler, A., Brown, C.: Explanation and limitations of study plot stability indices for forecasting dry snow slab avalanches in surrounding terrain, Cold Regions Science and Technology, 50(1-3), 23-34, https://doi.org/10.1016/j.coldregions.2007.02.010, 2007.

- Slaughter, A. E.: Numerical analysis of conditions necessary for near-surface snow metamorphism, PhD thesis, Montana State University, Bozeman, Montana, USA, 2010. https://search.proquest.com/openview/4e7f8f2f70589efc81d6d9198d67ee62/1?pq-origsite=gscholarcbl=18750diss=y

---

## Referee Comment (RC2) · Anonymous Referee #2 · 25 Apr 2020

In their manuscript "Sensitivity of modeled snow stability data to meteorological input uncertainty", the authors perform a sensitivity analysis of modeled snow stability data and indices to uncertainties in the meteorological forcing data. For this purpose, the widely used snow cover model SNOWPACK is forced with disturbed meteorological input data implementing different bias scenarios on the single meteorological parameters resulting in14000 simulations.

General Comments

The manuscript is very well written, and it presents valuable insight in snow stability modeling and its sensitivity to meteorological forcing data. Besides some minor issues, the presented methods are well explained and the manuscript fits well in the scope of NHESS. It represents an important and profound step towards more knowledge and

trust in using snow cover models in operational avalanche forecasting. I list some minor general comments and suggestions and specific remarks in the following.

I understand that this is a model sensitivity study and the model has been validated in other studies. However, I would highly appreciate if you could add some model validation for your presented case study to get a better understanding of the model performance especially with respect to the model's sensitivity to forcing errors. As I understand, you have some observed profiles available, maybe directly at the WFJ site? You could add a validation plot in Sect. 2.2 (e.g. accompanying Fig. 1?) for the undisturbed reference run after averaging the SNOWPACK layers as described there. I see that you perform kind of validation by comparing the results to avalanche activity and AAI, but it would be very valuable to have a direct comparison to measurements, in the best case even within the uncertainty range figures (Figs. 3 and 5). In addition, you should add modeled snow depth from the reference run to Fig. 2 (which I assume is observed snow depth, information should be added to the Fig. caption). All this would bring the findings of the impacts of forcing uncertainty on modeled snow stability in better context to reality and build more trust in the models to be used in operational forecasting.

I think the bias/disturbing procedure to produce the disturbed meteorological forcings within the given ranges needs some more explanation. Specifically: at what time scale are the errors applied? Is it a constant offset applied to the time series for a scenario or does it have some time variability within the scenario? This should then be referred to in L. 301-306.

Specific Comments

At some points in the manuscript you use "snow height", but mostly "snow depth". Please use "snow depth" consistently.

L. 15: ". . . sensitive to precipitation. . ."

L. 55: You state: "However, only a few studies have so far assessed the uncertainty of snow cover models." I would rather change this to, e.g., "However, only a few studies have so far assessed the impact of forcing uncertainty on the performance of snow cover models." because there are many studies available in literature which assess the performance and uncertainty of snow cover models in general.

L.105: "For the sensitivity analysis, we introduced uncertainties to the meteorological input." This sentence could be removed here, as you explain this in the next sections.

L. 150: I suggest to remove the sentence "For each scenario, 14,000 simulations were performed." here, as the number of simulations is explained in the following section 2.4. You could instead extend the last sentence of 2.4 (L. 170), e.g. like ". . . for each of the three applied scenarios."

L. 274 "Precipitation influences weak layer and slab properties." instead of "Precipitation influences weak and slab properties."

---

## Author Comment (AC1) · 14 Jul 2020

**Reply to Referee 2**

We thank the referee for the positive and very constructive feedback. In the following we will reply point-by-point. Your comments are in blue, replies in black.

In their manuscript "Sensitivity of modeled snow stability data to meteorological input uncertainty", the authors perform a sensitivity analysis of modeled snow stability data and indices to uncertainties in the meteorological forcing data. For this purpose, the widely used snow cover model SNOWPACK is forced with disturbed meteorological input data implementing different bias scenarios on the single meteorological parameters resulting in 14,000 simulations.

[Figure]

**General comments:**

I understand that this is a model sensitivity study and the model has been validated in other studies. However, I would highly appreciate if you could add some model validation for your presented case study to get a better understanding of the model performance especially with respect to the model's sensitivity to forcing errors. As I understand, you have some observed profiles available, maybe directly at the WFJ site? You could add a validation plot in Sect. 2.2 (e.g. accompanying Fig. 1?) for the undisturbed reference run after averaging the SNOWPACK layers as described there. I see that you perform kind of validation by comparing the results to avalanche activity and AAI, but it would be very valuable to have a direct comparison to measurements, in the best case even within the uncertainty range figures (Figs. 3 and 5). In addition, you should add modeled snow depth from the reference run to Fig. 2 (which I assume is observed snow depth, information should be added to the Fig. caption). All this would bring the findings of the impacts of forcing uncertainty on modeled snow stability in better context to reality and build more trust in the models to be used in operational forecasting.

As suggested, we will add the observed snow profiles to Figure 1. We will then present weak layer and slab layers with observed snow stratigraphy and the reference run in section 2.2. Furthermore, we will add modeled snow depth from the reference run to Figure 2. As we do not have manually observed snow profiles at the dates, we present in this study, we cannot show observed weak layer properties. Furthermore, this would need considerably more details on how slab and weak layer properties were presented from manual snow profiles (e.g. grain size 1 is defined by average grain size of all crystals and grain size 2 is defined by average grain size of largest crystals), so we prefer not add manual data to Figures 3 and 5.

I think the bias/disturbing procedure to produce the disturbed meteorological forcings within the given ranges needs some more explanation. Specifically: at what time scale are the errors applied? Is it a constant offset applied to the time series for a scenario

or does it have some time variability within the scenario? This should then be referred to in L. 301-306.

We will explain, how uncertainties are applied to the input in more detail. We will explicitly mention, that a bias b was randomly chosen for each variable and then that single value applied to the variable for a given time series. Furthermore, we will explicitly mention, that the given time series ranged from 1 October 2016 to 2 January 2017 for case WL, 3 January 2017 to 1 May 2017 for case SL and the entire season for case ALL.

**Specific Comments:**

At some points in the manuscript you use "snow height", but mostly "snow depth". Please use "snow depth" consistently.

For more consistency, we will change snow height to snow depth throughout the manuscript as suggested.

L. 15: "...sensitive **to** precipitation..."

We will change as suggested.

L. 55: You state: "However, only a few studies have so far assessed the uncertainty of snow cover models." I would rather change this to, e.g., "However, only a few studies have so far assessed the impact of forcing uncertainty on the performance of snow cover models." because there are many studies available in literature which assess the performance and uncertainty of snow cover models in general.

We will change as suggested.

L.105: "For the sensitivity analysis, we introduced uncertainties to the meteorological input." This sentence could be removed here, as you explain this in the next sections.

We will remove this sentence as suggested.

L. 150: I suggest to remove the sentence "For each scenario, 14,000 simulations were performed." here, as the number of simulations is explained in the following section 2.4. You could instead extend the last sentence of 2.4 (L. 170), e.g. like "...for each of the three applied scenarios."

As suggested, we will move the content of this sentence to the next section.

L. 274 "Precipitation influences weak layer and slab properties." instead of "Precipitation influences weak and slab properties."

We will change as suggested.
* * *

---

## Author Comment (AC2) · 14 Jul 2020

**Reply to Referee 1 (Simon Horton)**

We thank Simon Horton for the positive and very constructive feedback. In the following we will reply to the comments point-by-point. Your comments are in blue, replies in black.

**General comments:**

This paper investigates how stability indices predicted by snowpack models are impacted by uncertainties in the weather inputs. Spatial snowpack simulations could be valuable for avalanche forecasting, however there are numerous challenges in producing accurate spatially distributed weather inputs for these models. This paper provides

a strong quantitative analysis of what the implication of these uncertainties are when assessing snowpack stability. The sensitivity analysis uses weather data for a situation where a prominent weak layer formed in the snowpack and subsequently resulted in avalanche activity throughout the season. Although only a single scenario is investigated, the implications of various biases added into the data provides a robust analysis of how uncertainties in different weather inputs impacts weak layer formation, slab properties, and snowpack stability. The paper provides a significant contribution by improving the interpreting stability indices and illustrating the need for improved prediction of snowfall patterns. The methods are valid and rigorous, and the manuscript is well structured, organized, and easy to follow. My comments are relatively minor and could improve the manuscript by clarifying a few details and expanding on some interesting results.

**Specific comments:**

An interesting result that could use more discussion is explaining why the uncertainties resulted in unequal proportions of properties relative to the reference run. For example, Fig. 3 shows the majority cases had weak layers with lower densities and larger grain sizes than the reference run, and Fig. 9 shows the majority of cases had smaller critical crack lengths than the reference run. While such trends are reported throughout the results, they are not explained in the Discussion. Do these results mean (a) the distribution of input uncertainties were biases towards these results, (b) there were interaction effects between different combinations of biases that favoured these results, (c) some type of non-linearities in the model, (d) something else? If related to the biases, which biases resulted in these trends and why?

The results shown in Figure 3 likely resulted from the lognormal bias distribution in P, resulting in more runs with lower precipitation than the reference run (see Figure 1 in this Reply). We will address this in the Discussion section: "Introducing a lognormal distribution for the bias in precipitation resulted in unequal proportions relative to the reference run. A coefficient of variation for the lognormal distribution was chosen as

this reflects typical snow depth patterns observed in mountainous terrain (e.g. Liston, 2004). Hence, relatively more simulations had smaller P values than the reference run. As thinner snow covers generally have a lower density (less settlement) and experience larger temperature gradients, weak layer density decreased and grain size increased with decreasing precipitation (Figure 3b and c)."

There could be a bit more clarity on how the biases were applied to the weather data, since the distribution of weather inputs has a substantial effect on the results. I interpreted the method as follows: for a given time series, a bias b was randomly chosen for each variable and then that single value applied to the variable for the entire season. This could be stated more explicitly. If random biases were selected for each variable you would expect a roughly equal proportions of different bias combinations (e.g. samples with P+/TA+, P+/TA-, P-/TA+, P-/TA-). Would such combinations reflect the distribution of conditions you would actually expect to find in nature? Is this method consistent with other sensitivity studies using weather data? I suspect the method of applying these biases resulted in the skewed proportions discussed in the previous comment.

Indeed, your interpretation of how we applied the biases is correct. We will mention this more explicitly in the revised manuscript. Biases were applied randomly to each variably and independently of other variables. As such, we did not account for correlations between variables typically observed in nature. Nevertheless, the Sobol' method is advantageous in that it is model independent, can handle non-linear systems, and is among the most robust sensitivity methods (Saltelli and Annoni, 2010; Saltelli, 1999). The skewed proportion from the previous comment likely come from the lognormal distribution of bias introduced for P (see answer above) and not from the combination of different biases.

A limitation of the study is that it considers a single type of weak layer and snowpack structure combinations (i.e. early season facets above a crust). The type of weak layer considered in this study is important and should be stated in more places (e.g. abstract

and conclusions). While briefly discussed in lines 317-323, many of the results likely still generalize to more types of snowpack conditions (especially the slab properties). For surface hoar, a major sensitivity is the exposure time of the layer on the surface in between precipitation events. A light amount of snow could stop surface hoar growth in a much more dramatic way than facets. This again strengthens the argument that precipitation patterns (spatial, quantity, and timing!) are critical. While the details of surface hoar formation are outside the scope of this study, acknowledgement of this limitation and more discussion of what results likely transfer to other weak layers would be valuable.

Thanks for the suggestion. We will mention the type of weak layer explicitly in the Abstract: "Simulations were performed for a winter season, which was marked by a prolonged dry period at the beginning of the season. During this period, the snow surface layers transformed into faceted and depth hoar crystals, which were subsequently buried by snow. The early season snow surface was likely the weak layer of many avalanches later in the season." We will also mention the type of WL explicitly in the Conclusions: "We investigated the sensitivity of two modeled snow instability metrics for a weak layer consisting of faceted and depth hoar crystals..." We agree that our results for the slab properties are indeed more transferable to other types of weak layers. However, since we already discussed this in lines 336-342, we do not feel that this needs to be pointed out more prominently.

While the paper touches on most of the interesting results, there are a few minor results listed in the Technical comments that could also be discussed (e.g. why does wind speed impact shear strength?, why does weak layer grain size on 2 Jan not show sensitivity to temperature or radiation as might be expected for facets?)

We will add more discussion as suggested (see answers below to technical comments).

The discussion section could be reduced as there is substantial repetition from previous sections (e.g. lines 281-282 repeat the methods, lines 301-306 repeat introduction/motivation of study, lines 307-309 repeat methods, etc.). While this section is well written and examines interesting results, the repetition of why and how the study was done is unnecessary.

We will to remove redundant passages in the Discussion section as suggested.

The conclusions could have greater a emphasis on the contributions of the study. Although well written, they primarily focus is summarizing the results.

We agree and will put more emphasis on the contributions.

Overall the figures are clear, legible, and are effective at communicating the key results of the study.

**Technical comments:**

p1 l8: It would be helpful for the abstract to briefly explain the snowpack conditions for the case study (especially the fact the type of weak layer was early season facets above a crust).

We will mention the snowpack conditions in the Abstract as suggested, by adding: "Simulations were performed for a winter season, which was marked by a prolonged dry period at the beginning of the season. During this period, the snow surface layers transformed into faceted and depth hoar crystals, which were subsequently buried by snow. The early season snow surface was likely the weak layer of many avalanches later in the season.

p1 l 17: add "(more stable)" following crack length sentence for consistent structure.

We will change as suggested.

p1 l15 "sensitive **to** precipitation"

We will change as suggested.

p2 l49-52: It would be helpful to explicitly explain how to interpret SK38 and rc in

relation to initiation and propagation (e.g. "low values of SK38 indicate initiation more likely, low values of rc indicate propagation more likely")

Thanks for suggestion. We will explain the instability metrics in more detail in the revised manuscript.

p2-3 l 53-74: I appreciate how this paragraph concludes by identifying the clear gap in literature that this study addresses, however most of the paragraph reads like a long list of studies and the link to your research question isn't always apparent. I think by rewording some sentences it could be clearer how these studies relate to your research question. Also, Andrew Slaughter's PhD thesis (Slaughter, 2010) performs a SOBOL sensitivity analysis for formation of several types of weak layers and is relevant to this study.

As suggested, we will rewrite this section to make it more focused. Thanks for pointing out the work by Slaughter, which is indeed relevant to this study.

p4 l1: Just a comment: the weekly snow profiles aren't directly used in your study, although I assume they were important for understanding the avalanche conditions that you describe.

We will add the weekly snow profiles to facilitate validation of the model runs as suggested by referee 2.

p4 l115: Thickness-weighted averaging may smooth out the properties of the most unstable layer(s) that may contain the critical properties for avalanche release. Could this averaging method somehow impact the biases favouring the formation of more unstable layers?

We agree that using thickness-weighted averaging may smooth out properties of the most unstable layers. However, since there is no unambiguous definitions of the most unstable layer, as layers with a lowest rc value do not necessarily have the lowest SK38 value, we decided to focus on average properties. Note that initially we also tried to

focus on single layers, rather than average properties, and the overall observed trends were very similar. We do not believe, that averaging layer properties favours more unstable layers (see answer to first comment above).

p4 l116: "shear strength of the weak layer. . ."

We will change as suggested.

p4 l116: I understand you present the SK38 and rc derivations in general form, but would it make sense to use the bar notation for the variables that you substitute with thickness-weighted averages (such as slab and weak layer densities)?

Thanks for noting this unclear explanation in the text. The instability metrics were calculated for each layer, as presented in the manuscript, and then thickness-weighted average instability metrics were reported. We will clarify this in the manuscript, by using the bar notation more consistently throughout the manuscript as suggested and adding: "Then, we determined the thickness-weighted mean instability metrics $\overline{\text{SK38}}$ and $\overline{r_c}$, which are reported in the following."

p5 l131-134: Please provide a written explanation of what this correction factor accounts for.

We will introduce the correction factor providing the following explanation: "Richter et al. (2019) introduced the correction factor $F_{wl}$ to replace two variables of the original parameterization (Gaume et al., 2017), which were not well defined in SNOWPACK. The factor $F_{wl}$ accounts for weak layer density and grain size and considerably improved the $r_c$ parameterization, and it ensures that layers with larger grains have lower $r_c$ values (Richter et al., 2019)."

p5 l 136: In the abstract you specify the uncertainty values are typical for extents of 2 km and elevation changes of 200 m. It would be worth including that somewhere in the text.

We will include the interpretation of uncertainties at the end of this paragraph: "With

the given ranges and distributions (Table 1), biases can be interpreted as differences typically observed within a spatial distance of around 2 km and an elevation range of around 200 m. For example, around 68 % of the simulations have a bias in air temperature of -1 K to +1 K, which cover temperature differences within an elevation band of around 200 m. Uncertainties in P will yield rather shallow or rather thick snowpacks as typically observed for wind exposed or wind sheltered slopes."

p6 l150: Please specify here whether Case ALL has a unique set of biases or simply concatenates the two other cases.

We will state that for each scenario (case WL, case SL, and case ALL) a unique set of biases was introduced.

Sect. 2.4: This section could use some additional explanation. First, it would be helpful to move the written description of what $ST_i$ means (line 160-161) before the mathematical definition in Eq. 5. On line 162 you describe a 'perfect additive model', but do not explain whether this is important or how that idea applies to this study. It's not clear what information is contained in the A and B matrices as you simply describe their dimensions rather than their content, and thus the importance of AB is unclear. Without explanation I'm wondering if B is a matrix full of biases b you introduce in Table 1 (i.e. the same letter).

We will improve the clarity of the explanation in this section. First, we will move lines 160-162 above equation (5) and remove the reference to a perfect additive model, as it does not apply to our case: "In a global sensitivity analysis, the total-order sensitivity index $ST_i$ describes the variance in output variables Y, i.e. snow properties, due to uncertainties introduced to a specific meteorological input $X_i$, while including interactions with other forcing errors. Values for $ST_i$ range from 0 (no sensitivity) to 1 (one-to-one sensitivity)." We will also better explain the content of the matrices A and B, by adding this sentence: "The elements of the two independent matrices A and B thus consist of biases for the input variables randomly picked from the ranges and distributions shown

in Table 1."

Sect. 3.1: This section provides a very clear and helpful practical explanation of the case study.

Thank you very much for this feedback.

p8 l195: A more intuitive wording would be something like "We present weak layer and slab properties on 2 January with results from the reference run and case WL because..."

We will change as suggested.

p8 l197: In the methods your weak layer group consists of more than just facets (e.g depth hoar and surface hoar), does the "percent facets" variable actually mean percent of weak layers or literally percent facets and there was no depth hoar or surface hoar?

Indeed, this variable was unclear and we will define it more clearly in the Methods section (p.4, l.115): "Next to weak layer properties, we investigated the percentage of facets (% facets), which was defined as the sum of thicknesses of all layers consisting of either facets, depth hoar or surface hoar crystals, divided by the sum of thicknesses of all layers deposited between these two dates (see Section 3.2.1)."

Fig. 3 and 4: Would a more logical progression be showing Fig. 4 first to show which input uncertainties had the greatest effect then show Fig. 3 to show the direction of the effect? Seeing which weather input had the greatest impact on a given property would help explain why a specific scatter plot is being shown. Same logic applies to Fig. 5 and 6 and 9 and 10. Just a thought.

We will change the order of the Figures and the corresponding text as suggested.

p9 l204: Is weak layer thickness also calculated as an average of each individual layer, or was it the sum of all identified weak layers? The sum seems more meaningful.

The weak layer thickness was calculated as the sum of all identified weak layers.
Thanks for noting this unclear definition. We will clarify this by adding to section 2.2: "The only exception were weak layer thickness and slab thickness, that were obtained by adding the thicknesses of all weak layers and slab layers, respectively."

p9 l204-205: This result about the impact of precipitation is somewhat unique to how this weak layer is being identified (as all layers forming over a date range), and it is not necessarily intuitive to think about how precipitation during a formation period impacts weak layer formation. It would be helpful to reiterate what precipitation means for this specific case. Also, wouldn't you expect grain size to be more sensitive to air temperature (and perhaps the radiation variables) given the weak layers were faceted crystals?

We agree that repeating the meaning of precipitation would be helpful for interpretation, so we will mention this in the text: "Increasing P led to denser weak layers and smaller grains (Figure 4b,c). Here, we want to recall that positive biases in P result in thicker snowpacks, as would typically be observed in wind-sheltered locations." Regarding the sensitivity of weak layer grain size to air temperature and radiation, this result is indeed somewhat surprising, since both these parameters are highly relevant for the energy input at the snow surface and thus snow surface temperature and temperature gradients across the snowpack. However, in December, the energy balance at the snow surface is generally negative (i.e. surface cooling), as days are very short and incoming short-wave radiation is very low. Even with positive air temperature, the snow surface often stays well below zero, except on very steep south-facing slopes (higher incoming short-wave radiation), or when there is a thick cloud cover (higher incoming long-wave radiation). Since there was generally only limited cloud cover in December 2016 (low incoming long-wave radiation), and the simulations were performed for a flat field site (low incoming short-wave radiation), we believe our results are plausible.

p9 l208: It would be interesting to discuss why weak layer shear strength was most sensitive to wind speed as well as the direction of the relationship (i.e. did increasing wind typically result in higher or lower shear strength?). This result is not necessarily

the most intuitive and could be discussed more.

The weak layer consisted of layers deposited between two given dates and consisting of persistent grain types (i.e. DH, FC or SH). Shear strength in SNOWPACK is a function of grain type and density. As new snow density in SNOWPACK depends on wind velocity, we believe that shear strength depended on wind velocity for the case WL. A secondary effect could be the destruction of surface hoar due to wind. Indeed, layers of SH on the snow surface will be destroyed if the wind speed exceeds a certain threshold value in SNOWPACK. Therefore, we expect fewer SH with increasing wind speed. We will discuss this in more detail.

p11 l211: It would be helpful to introduce this date the same way as 2 Jan by introducing the fact you now consider all three cases before you start reporting results.

We will introduce the date as suggested: "According to Section 3.2.1, we present weak layer and slab properties on 9 March 2017 by considering all three cases with results from the reference run."

Fig. 5: The load-P subplots present obvious results and it's not clear there's added value in graphing these relationships.

We agree that the subplots for the load are rather trivial. Nevertheless, we intentionally added these figures to highlight that although load was most sensitive to precipitation in all three scenarios, for case WL the variability in slab load was almost imperceptible.

p12 l223: High slab load than what? The reference case?

That is correct. We will add that slab load was higher than in the reference run.

p12 l227: Does it make sense that ST would change between 2 Jan and 9 mar if Case WL uses the reference data from 2 Jan onwards?

You are correct to assume that ST does not change much between 2 January and 9 March. Nevertheless, there are some subtle changes, as different weak layers on 2

January do not necessarily react exactly the same to the same slab. Indeed, harder and denser weak layers will settle less than soft low density weak layers. As such, there are some changes in ST between 2 January and 9 March.

p12 l233-243: This paragraph is very well written and easy to follow!

Thanks.

Fig. 8: Is it correct to follow the points as a time series starting from the bottom left? If so, including a line connecting the points (and perhaps even an arrow) could make it clearer this shows evolving stability properties rather than an independent scatter of data points.

Indeed, this is the case. We will improve as suggested.

p13 l 254: Could you provide a similar summary for rc as done for SK38 in line 247 ("This suggests, that different slabs influenced SK38 more than different weak layers"). It appears from Fig. 9 rc was equally impacted by weak layer and slab properties.

We agree with your interpretation and will provide this summary: "This suggests that rc was equally impacted by weak layer and slab properties."

p14 l259-268: I found this paragraph slightly confusing to read. Perhaps some parts could be reworded or even some of the interpretation moved to the Discussion.

We will improve the paragraph as suggested.

p18 l320-324: This result agrees with Horton et al. (2015) who examine how variability in meteorological fields from NWP models across elevations resulted in reasonable predictions of surface hoar formation. Slaughter (2010) also analyzes sensitives of surface hoar and other weak layers to weather inputs.

Thanks for this input, we will refer to these studies in the revised manuscript: "This result agrees with Horton et al. (2015) who examined how variability in meteorological fields from numerical weather prediction models across elevations resulted in reasonable predictions of surface hoar formation. However, we only looked at one type of weak layer. The formation and subsequent burial of surface hoar might be more sensitive to other meteorological parameters, such as wind speed (Stössel et al., 2010). In fact, Slaughter (2010) investigated the sensitivity of near-surface faceting and surface hoar formation at mid-day and mid-night to input parameters using a snow thermal model. He found incoming long-wave radiation to be the most dominant input parameter, although they did not investigate the sensitivity to precipitation."

p18 l340-342: This is a very practical take away from this study that supports practical forecasting experience, and could be a valuable application of snowpack models.

Thanks. As suggested above, we will add this as an application to the Conclusions.

p18 l343: "than in weak layer shear strength"

We will change as suggested.

p18 l344-345: These results could be supported by citing field studies that describe the lag in weak layer shear strength increases after loading, such as Jamieson et al. (2007) who also give interesting implications on spatial variability of stability indexes due to variable precipitation.

We do not entirely agree that field observations of the lag in shear strength increase after loading support our results. In our case, we are discussing observed trends in SK38 in March, after several precipitation events and when the weak layers are already more than 60 days old. We will clarify this, by adding that during periods without precipitation, SK38 slightly increased, with can indeed be supported by Jamieson et al. (2007). Whereas, the overall decrease in SK38 (i.e. low values end of March) was explained by a stronger increase in slab load than in weak layer shear strength between January and March (i.e. shear strength remained low in March). The same effect resulted in a decrease in the stability index (e.g. SK38) with increasing P on 9 March 2017 (Figure 9c), i.e. slab load increased stronger than weak layer shear

strength with increasing P.

p18 lp351: How do you explain this counter intuitive result where SK38 remain slow into spring? It would seem that since the load continues to increase that the weak layer strength must have remained low. Was this the case?

After 10 March 2017, the slab load hardly increased, since there was almost no precipitation (see Figure 1). During the period of slab formation, the slab load increased considerably stronger than the weak layer shear strength, resulting in low values of SK38 in spring (see answer above). This is a well-known problem with SK38, and why it should not be used for weak layers that are buried deeper than about 100 cm (Schweizer et al., 2016).

p19 l360-361: How does this sentence about precipitation tie back to the theme of climate change?

We will add: "Furthermore, with climate change, extreme events may become more frequent, e.g. prolonged dry periods may remain - favoring the formation of weak layers - and may alternate with more extreme precipitation events (CH2018, 2018)."

**References:**

CH2018: CH2018 - Climate Scenarios for Switzerland, Technical Report, National Centre for Climate Services, Zurich, ISBN: 978-3-9525031-4-0, 2018.

Gaume, J., van Herwijnen, A., Chambon, G., Wever, N., and Schweizer, J.: Snow fracture in relation to slab avalanche release: critical state for the onset of crack propagation, The Cryosphere, 11, 217–228, 2017.

Horton, S., Schirmer, M., and Jamieson, B.: Meteorological, elevation, and slope effects on surface hoar formation, The Cryosphere, 9, 1523–1533, 2015.

Jamieson, B., Zeidler, A.,Brown, C.: Explanation and limitations of study plot stability indices for forecasting dry snow slab avalanches in surrounding terrain, Cold Regions

Science and Technology, 50(1-3), 23-34, 2007.

Liston, G. E.: Representing Subgrid Snow Cover Heterogeneities in Regional and Global Models, Journal of Climate, 17, 1381–1397, 2004.

Richter, B., Schweizer, J., Rotach, M. W., and van Herwijnen, A.: Validating modeled critical crack length for crack propagation in the snow cover model SNOWPACK, The Cryosphere, 13, 3353–3366, 2019.

Saltelli, A. and Annoni, P.: How to avoid a perfunctory sensitivity analysis, Environmental Modelling Software, 25, 1508–1517, 2010.

Saltelli, A., Annoni, P., Azzini, I., Campolongo, F., Ratto, M., and Tarantola, S.: Variance based sensitivity analysis of model output. Design and estimator for the total sensitivity index, Computer Physics Communications, 181, 259 – 270, 2010.

Schweizer, J., Reuter, B., van Herwijnen, A., Richter, B., and Gaume, J.: Temporal evolution of crack propagation propensity in snow in relation to slab and weak layer properties, The Cryosphere, 10, 2637–2653, 2016.

Slaughter, A.E.: Numerical analysis of conditions necessary for near-surface snow metamorphism, PhD thesis, Montana State University, Bozeman, Montana, USA, 2010.

Stössel, F., Guala, M., Fierz, C., Manes, C., and Lehning, M.: Micrometeorological and morphological observations of surface hoar dynamics on a mountain snow cover, Water Resources Research, 46, W04 511, 2010.

**Fig. 1.** Distribution of meteorological input uncertainties.

---

## Author Response (AR1)

**Reply to Referee #1 (Simon Horton)**

We thank Simon Horton for the positive and very constructive feedback. In the following we will reply to the comments point-by-point. Your comments are in blue, replies in black.

General comments:

This paper investigates how stability indices predicted by snowpack models are impacted by uncertainties in the weather inputs. Spatial snowpack simulations could be valuable for avalanche forecasting, however there are numerous challenges in producing accurate spatially distributed weather inputs for these models. This paper provides a strong quantitative analysis of what the implication of these uncertainties are when assessing snowpack stability. The sensitivity analysis uses weather data for a situation where a prominent weak layer formed in the snowpack and subsequently resulted in avalanche activity throughout the season. Although only a single scenario is investigated, the implications of various biases added into the data provides a robust analysis of how uncertainties in different weather inputs impacts weak layer formation, slab properties, and snowpack stability. The paper provides a significant contribution by improving the interpreting stability indices and illustrating the need for improved prediction of snowfall patterns. The methods are valid and rigorous, and the manuscript is well structured, organized, and easy to follow. My comments are relatively minor and could improve the manuscript by clarifying a few details and expanding on some interesting results.

Specific comments:

An interesting result that could use more discussion is explaining why the uncertainties resulted in unequal proportions of properties relative to the reference run. For example, Fig. 3 shows the majority cases had weak layers with lower densities and larger grain sizes than the reference run, and Fig. 9 shows the majority of cases had smaller critical crack lengths than the reference run. While such trends are reported throughout the results, they are not explained in the Discussion. Do these results mean (a) the distribution of input uncertainties were biases towards these results, (b) there

were interaction effects between different combinations of biases that favoured these results, (c) some type of non-linearities in the model, (d) something else? If related to the biases, which biases resulted in these trends and why?

The results shown in Figure 3 resulted from the lognormal bias distribution in P, resulting in more runs with lower precipitation than the reference run (see Fig 1 below). We addressed this in the Discussion section (p.16, l.300-303): "Introducing a lognormal distribution for the bias in precipitation resulted in unequal proportions relative to the reference run (e.g. Figures **??** and **??**). A coefficient of variation for the lognormal distribution was chosen as this reflects typical snow depth patterns observed in mountainous terrain (e.g. Liston, 2004). Hence, relatively more simulations had smaller P values than the reference run." Furthermore, we added in p.17,l.348-350: "As thinner snow covers generally have a lower density (less settlement) and experience larger temperature gradients, weak layer density decreased and grain size increased with decreasing precipitation (Figure 3b and c)."

[Figure]

Figure 1: Distribution of meteorological input uncertainties.

There could be a bit more clarity on how the biases were applied to the weather data, since the distribution of weather inputs has a substantial effect on the results. I interpreted the method as follows: for a given time series, a bias b was randomly chosen for each variable and then that single value applied to the variable for the entire season. This could be stated more explicitly. If random biases were selected for each variable you would expect a roughly equal proportions of different bias combinations (e.g. samples with P+/TA+, P+/TA-, P-/TA+, P-/TA-). Would such combinations reflect the distribution of conditions you would actually expect to find in nature? Is this method consistent with other sensitivity studies using weather data? I suspect the method of applying these biases resulted in the skewed proportions discussed in the previous comment.

Indeed, your interpretation of how we applied the biases is correct. We mentioned this more explicitly in the revised manuscript (p.6, l.148). Biases were applied randomly to each variably and independently of other variables. As such, we did not account for correlations between variables typically observed in nature. Nevertheless, the Sobol' method is advantageous in that it is model independent, can handle non-linear systems, and is among the most robust sensitivity methods (Saltelli and Annoni, 2010; Saltelli, 1999). The skewed proportion from the previous comment likely come from the lognormal distribution of bias introduced for P (see answer above) and not from the combination of different biases.

A limitation of the study is that it considers a single type of weak layer and snowpack structure combinations (i.e. early season facets above a crust). The type of weak layer considered in this study is important and should be stated in more places (e.g. abstract and conclusions). While briefly discussed in lines 317-323, many of the results likely still generalize to more types of snowpack conditions (especially the slab properties). For surface hoar, a major sensitivity is the exposure time of the layer on the surface in between precipitation events. A light amount of snow could stop surface hoar growth in a much more dramatic way than facets. This again strengthens the argument that precipitation patterns (spatial, quantity, and timing!) are critical. While the details of surface hoar formation are outside the scope of this study, acknowledgement of this limitation and more discussion of what results likely transfer to other weak layers would be valuable.

Thanks for the suggestion. We mentioned the type of weak layer explicitly in the Abstract (p.1, l.8): "Simulations were performed for a winter season, which was marked by a prolonged dry period at the beginning of the season. During this period, the snow surface layers transformed into faceted and depth hoar crystals, which were subsequently buried by snow. The early season snow surface was likely the weak layer of many avalanches later in the season." We will also mention the type of WL explicitly in the Conclusions: "We investigated the sensitivity of two modeled snow instability metrics for a weak layer consisting of faceted and depth hoar crystals..." We agree that our results for the slab properties are indeed more transferable to other types of weak layers. However, since we already discussed this in lines 336-342, we do not feel that this needs to be pointed out more prominently.

While the paper touches on most of the interesting results, there are a few minor results listed in the Technical comments that could also be discussed (e.g. why does wind speed impact shear strength?, why does weak layer grain size on 2 Jan not show sensitivity to temperature or radiation as might be expected for facets?)

We added more discussion as suggested in p.167,l.339-355 in the revised manuscript (see answers below to technical comments).

The discussion section could be reduced as there is substantial repetition from previous sections (e.g. lines 281-282 repeat the methods, lines 301-306 repeat introduction/motivation of study, lines 307-309 repeat methods, etc.). While this section is well written and examines interesting results, the repetition of why and how the study was done is unnecessary.

We removed redundant passages in the Discussion section as suggested.

The conclusions could have greater a emphasis on the contributions of the study. Although well written, they primarily focus is summarizing the results.

We agree and put more emphasis on the contributions (p.19, l. 4066-409 and l. 415-416).

Overall the figures are clear, legible, and are effective at communicating the key results of the study.

Technical comments:

p1 l8: It would be helpful for the abstract to briefly explain the snowpack conditions for the case study (especially the fact the type of weak layer was early season facets above a crust).

We mentioned the snowpack conditions in the Abstract as suggested (p.1, l.7-9), by adding: "Simulations were performed for a winter season, which was marked by a prolonged dry period at the beginning of the season. During this period, the snow surface layers transformed into faceted and depth hoar crystals, which were subsequently buried by snow. The early season snow surface was likely the weak layer of many avalanches later in the season."

p1 l 17: add "(more stable)" following crack length sentence for consistent structure.

We changed as suggested.

p1 l15 "sensitive to precipitation"

We changed as suggested.

p2 l49-52: It would be helpful to explicitly explain how to interpret SK38 and rc in relation to initiation and propagation (e.g. "low values of SK38 indicate initiation more likely, low values of rc indicate propagation more likely")

Thanks for suggestion. We explained the instability metrics in more detail in the revised manuscript (p.2,l.52).

p2-3 l 53-74: I appreciate how this paragraph concludes by identifying the clear gap in literature that this study addresses, however most of the paragraph reads like a long list of studies and the link to your research question isn't always apparent. I think by rewording some sentences it could be clearer how these studies relate to your research question. Also, Andrew Slaughter's PhD thesis (Slaughter, 2010) performs a SOBOL sensitivity analysis for formation of several types of weak layers and is relevant to this study.

As suggested, we rewrote this section (p.2, l.53 - p.3, l.69) to make it more focused. Thanks for pointing out the work by Slaughter, which is indeed relevant to this study.

p4 l1: Just a comment: the weekly snow profiles aren't directly used in your study, although I assume they were important for understanding the avalanche conditions that you describe.

We added the weekly snow profiles to Figure 1 to facilitate validation of the model runs as suggested by referee 2.

p4 l115: Thickness-weighted averaging may smooth out the properties of the most unstable layer(s) that may contain the critical properties for avalanche release. Could this averaging method somehow impact the biases favouring the formation of more unstable layers?

We agree that using thickness-weighted averaging may smooth out properties of the most unstable layers. However, since there is no unambiguous definitions of the most unstable layer, as layers with a lowest rc value do not necessarily have the lowest SK38 value, we decided to focus on average properties. Note that initially we also tried to focus on single layers, rather than average properties, and the overall observed trends were very similar. We do not believe, that averaging layer properties favours more unstable layers (see answer to first comment above).

We changed as suggested.

p4 l116: I understand you present the SK38 and rc derivations in general form, but would it make sense to use the bar notation for the variables that you substitute with thickness-weighted averages (such as slab and weak layer densities)?

Thanks for noting this unclear explanation in the text. The instability metrics were calculated for each layer, as presented in the manuscript, and then thickness-weighted average instability metrics were reported. We clarified this in the manuscript, by using the bar notation more consistently throughout the manuscript as suggested and adding (p.6, l. 142): "SK38 an $r_c$ were calculated for each of the weak layers as defined above and thickness-weighted mean instability metrics $\overline{SK38}$ and $\overline{r_c}$ were determined from all weak layers"

p5 l131-134: Please provide a written explanation of what this correction factor accounts for.

We introduced the correction factor providing the following explanation (p.6, l.137): "Richter et al. (2019) introduced the correction factor Fwl to replace two variables of the original parameterization (Gaume et al., 2017), which were not well defined in SNOWPACK. The factor Fwl accounts for weak layer density and grain size and considerably improved the rc parameterization, and it ensures that layers with larger grains have lower rc values (Richter et al., 2019)."

p5 l 136: In the abstract you specify the uncertainty values are typical for extents of 2 km and elevation changes of 200 m. It would be worth including that somewhere in the text.

We included the interpretation of uncertainties at the end of this paragraph (p.6, l.155): "With the given ranges and distributions (Table 1), biases can be interpreted as differences typically observed within a spatial distance of around 2 km and an elevation range of around 200 m. For example, around 68 % of the simulations have a bias in air temperature of -1 K to +1 K, which cover temperature differences within an elevation band of around 200 m. Uncertainties in P will yield rather shallow or rather thick snowpacks as typically observed for wind exposed or wind sheltered slopes."

p6 l150: Please specify here whether Case ALL has a unique set of biases or simply concatenates the two other cases.

We stated that for each scenario (case WL, case SL, and case ALL) a unique set of biases was introduced (p.7, l. 167).

Sect. 2.4: This section could use some additional explanation. First, it would be helpful to move the written description of what $ST_i$ means (line 160-161) before the mathematical definition in Eq. 5. On line 162 you describe a 'perfect additive model', but do not explain whether this is important or how

that idea applies to this study. It's not clear what information is contained in the A and B matrices as you simply describe their dimensions rather than their content, and thus the importance of AB is unclear. Without explanation I'm wondering if B is a matrix full of biases b you introduce in Table 1 (i.e. the same letter).

We improved the clarity of the explanation in this section. First, we moved lines 160-162 above equation (5) and removed the reference to a perfect additive model, as it does not apply to our case (p.7, l.173): "In a global sensitivity analysis, the total-order sensitivity index STi describes the variance in output variables Y, i.e. snow properties, due to uncertainties introduced to a specific meteorological input Xi, while including interactions with other forcing errors. Values for STi range from 0 (no sensitivity) to 1 (one-to-one sensitivity)." We also better explained the content of the matrices A and B, by adding this sentence (p.7., l.179): "The elements of the two independent matrices A and B thus consist of biases for the input variables randomly picked from the ranges and distributions shown in Table 1."

Sect. 3.1: This section provides a very clear and helpful practical explanation of the case study.

Thank you very much for this feedback.

p8 l195: A more intuitive wording would be something like "We present weak layer and slab properties on 2 January with results from the reference run and case WL because..."

We changed as suggested.

p8 l197: In the methods your weak layer group consists of more than just facets (e.g depth hoar and surface hoar), does the "percent facets" variable actually mean percent of weak layers or literally percent facets and there was no depth hoar or surface hoar?

Indeed, this variable was unclear and we defined it more clearly in the Methods section (p.5, l.112): "Hence, weak layer thickness $D_{wl}$ was defined as the thickness of all layers consisting of either facets, depth hoar or surface hoar, which were deposited between these two dates. The percentage of facets (%

facets), was defined as $D_{wl}$, divided by the total thickness of all layers which were deposited between these two dates (see Section 3.2.1)."

Fig. 3 and 4: Would a more logical progression be showing Fig. 4 first to show which input uncertainties had the greatest effect then show Fig. 3 to show the direction of the effect? Seeing which weather input had the greatest impact on a given property would help explain why a specific scatter plot is being shown. Same logic applies to Fig. 5 and 6 and 9 and 10. Just a thought.

We changed the order of the Figures and the corresponding text as suggested.

p9 l204: Is weak layer thickness also calculated as an average of each individual layer, or was it the sum of all identified weak layers? The sum seems more meaningful.

The weak layer thickness was calculated as the sum of all identified weak layers. Thanks for noting this unclear definition. We will clarify this by adding (p.5, l.112): "Hence, weak layer thickness $D_{wl}$ was defined as the thickness of all layers consisting of either facets, depth hoar or surface hoar, which were deposited between these two dates."

p9 l204-205: This result about the impact of precipitation is somewhat unique to how this weak layer is being identified (as all layers forming over a date range), and it is not necessarily intuitive to think about how precipitation during a formation period impacts weak layer formation. It would be helpful to reiterate what precipitation means for this specific case. Also, wouldn't you expect grain size to be more sensitive to air temperature (and perhaps the radiation variables) given the weak layers were faceted crystals?

We agree that repeating the meaning of precipitation would be helpful for interpretation, so we mentioned this in the text (p.10, l.233): "Increasing P led to denser weak layers and smaller grains (Figure 4b,c). Positive biases in P result in thicker snowpacks, as would typically be observed in wind-sheltered locations." Regarding the sensitivity of weak layer grain size to air temperature and radiation, this result is indeed somewhat surprising, since both these parameters are highly relevant for the energy input at the snow surface and thus snow surface temperature and temperature gradients across

the snowpack. However, in December, the energy balance at the snow surface is generally negative (i.e. surface cooling), as days are very short and incoming short-wave radiation is very low. Even with positive air temperature, the snow surface often stays well below zero, except on very steep south-facing slopes (higher incoming short-wave radiation), or when there is a thick cloud cover (higher incoming long-wave radiation). Since there was generally only limited cloud cover in December 2016 (low incoming long-wave radiation), and the simulations were performed for a flat field site (low incoming short-wave radiation), we believe our results are plausible. We discussed this in more detail in p.17, l.340-348 in the revised manuscript.

p9 l208: It would be interesting to discuss why weak layer shear strength was most sensitive to wind speed as well as the direction of the relationship (i.e. did increasing wind typically result in higher or lower shear strength?). This result is not necessarily the most intuitive and could be discussed more.

The weak layer consisted of layers deposited between two given dates and consisting of persistent grain types (i.e. DH, FC or SH). Shear strength in SNOWPACK is a function of grain type and density. As new snow density in SNOWPACK depends on wind velocity, we believe that shear strength depended on wind velocity for the case WL. We discussed this in more detail in p.17, l.350 in the revised manuscript.

p11 l211: It would be helpful to introduce this date the same way as 2 Jan by introducing the fact you now consider all three cases before you start reporting results.

We introduced the date as suggested: "According to Section 3.2.1, we present weak layer and slab properties on 9 March 2017 by considering all three cases with results from the reference run."

Fig. 5: The load-P subplots present obvious results and it's not clear there's added value in graphing these relationships.

We agree that the subplots for the load are rather trivial. Nevertheless, we intentionally added these figures to highlight that although load was most sensitive to precipitation in all three scenarios, for case WL the variability in slab load was almost imperceptible.

That is correct. We added that slab load was higher than in the reference run.

p12 l227: Does it make sense that ST would change between 2 Jan and 9 mar if Case WL uses the reference data from 2 Jan onwards?

You are correct to assume that ST does not change much between 2 January and 9 March. Nevertheless, there are some subtle changes, as different weak layers on 2 January do not necessarily react exactly the same to the same slab. Indeed, harder and denser weak layers will settle less than soft low density weak layers. As such, there are some changes in ST between 2 January and 9 March. We discussed this in p.18, l.353.

p12 l233-243: This paragraph is very well written and easy to follow!

Thanks.

Fig. 8: Is it correct to follow the points as a time series starting from the bottom left? If so, including a line connecting the points (and perhaps even an arrow) could make it clearer this shows evolving stability properties rather than an independent scatter of data points.

Indeed, this is the case. We improved as suggested.

p13 l 254: Could you provide a similar summary for rc as done for SK38 in line 247 ("This suggests, that different slabs influenced SK38 more than different weak layers"). It appears from Fig. 9 rc was equally impacted by weak layer and slab properties.

We agree with your interpretation and provided this summary (p.15, l.282): "This suggests that rc was equally impacted by weak layer and slab properties."

p14 l259-268: I found this paragraph slightly confusing to read. Perhaps some parts could be reworded or even some of the interpretation moved to the Discussion.

We improved the paragraph as suggested.

p18 l320-324: This result agrees with Horton et al. (2015) who examine how variability in meteorological fields from NWP models across elevations resulted in reasonable predictions of surface hoar formation. Slaughter (2010) also analyzes sensitives of surface hoar and other weak layers to weather inputs.

Thanks for this input, we referd to these studies in the revised manuscript (p.17, l.333-338): "This result agrees with Horton et al. (2015) who examined how variability in meteorological fields from numerical weather prediction models across elevations resulted in reasonable predictions of surface hoar formation. However, we only looked at one type of weak layer. The formation and subsequent burial of surface hoar might be more sensitive to other meteorological parameters, such as wind speed (Stössel et al., 2009). In fact, Slaughter (2010) investigated the sensitivity of near-surface faceting and surface hoar formation at mid-day and mid-night to input parameters using a snow thermal model. He found incoming long-wave radiation to be the most dominant input parameter, although they did not investigate the sensitivity to precipitation."

p18 l340-342: This is a very practical take away from this study that supports practical forecasting experience, and could be a valuable application of snowpack models.

Thanks. As suggested above, we added this as an application to the Conclusions.

p18 l343: "than in weak layer shear strength"

We changed as suggested.

p18 l344-345: These results could be supported by citing field studies that describe the lag in weak layer shear strength increases after loading, such as Jamieson et al. (2007) who also give interesting implications on spatial variability of stability indexes due to variable precipitation.

We do not entirely agree that field observations of the lag in shear strength

increase after loading support our results. In our case, we are discussing observed trends in SK38 in March, after several precipitation events and when the weak layers are already more than 60 days old. We clarified this in p.18, l.374-379, by adding that during periods without precipitation, SK38 slightly increased due to the lagged increase in shear strength (Jamieson et al., 2007). Whereas, the overall decrease in SK38 (i.e. low values end of March) was explained by a stronger increase in slab load than in weak layer shear strength between January and March (i.e. shear strength remained low in March). The same effect resulted in a decrease in the stability index (e.g. SK38) with increasing P on 9 March 2017 (Figure 9c), i.e. slab load increased stronger than weak layer shear strength with increasing P.

p18 l351: How do you explain this counter intuitive result where SK38 remain slow into spring? It would seem that since the load continues to increase that the weak layer strength must have remained low. Was this the case?

After 10 March 2017, the slab load hardly increased, since there was almost no precipitation (see Figure 1). During the period of slab formation, the slab load increased considerably stronger than the weak layer shear strength, resulting in low values of SK38 in spring (see answer above). This is a well-known problem with SK38, and why it should not be used for weak layers that are buried deeper than about 100 cm (Schweizer et al., 2016).

p19 l360-361: How does this sentence about precipitation tie back to the theme of climate change?

We added in p.19, l.395: "With climate change, extreme events may become more frequent, e.g. prolonged dry periods - favoring the formation of weak layers - may alternate with more extreme precipitation events (CH2018, 2018) - with partly opposing effects on our snow instability metrics."

**Reply to Referee #2**

We thank the referee for the positive and very constructive feedback. In the following we will reply point-by-point. Your comments are in blue, replies in black.

In their manuscript "Sensitivity of modeled snow stability data to meteorological input uncertainty", the authors perform a sensitivity analysis of modeled snow stability data and indices to uncertainties in the meteorological forcing data. For this purpose, the widely used snow cover model SNOWPACK is forced with disturbed meteorological input data implementing different bias scenarios on the single meteorological parameters resulting in 14,000 simulations.

General comments:

I understand that this is a model sensitivity study and the model has been validated in other studies. However, I would highly appreciate if you could add some model validation for your presented case study to get a better understanding of the model performance especially with respect to the model's sensitivity to forcing errors. As I understand, you have some observed profiles available, maybe directly at the WFJ site? You could add a validation plot in Sect. 2.2 (e.g. accompanying Fig. 1?) for the undisturbed reference run after averaging the SNOWPACK layers as described there. I see that you perform kind of validation by comparing the results to avalanche activity and AAI, but it would be very valuable to have a direct comparison to measurements, in the best case even within the uncertainty range figures (Figs. 3 and 5). In addition, you should add modeled snow depth from the reference run to Fig. 2 (which I assume is observed snow depth, information should be added to the Fig. caption). All this would bring the findings of the impacts of forcing uncertainty on modeled snow stability in better context to reality and build more trust in the models to be used in operational forecasting.

As suggested, we added the observed snow profiles to Figure 1. We then presented weak layer and slab layers with observed snow stratigraphy and the reference run in section 2.2. Furthermore, we added modeled snow depth from the reference run to Figure 2. As we do not have manually observed snow profiles at the dates, we present in this study, we cannot show observed

weak layer properties. Furthermore, this would need considerably more details on how slab and weak layer properties were presented from manual snow profiles (e.g. grain size 1 is defined by average grain size of all crystals and grain size 2 is defined by average grain size of largest crystals), so we prefer not add manual data to Figures 3 and 5.

I think the bias/disturbing procedure to produce the disturbed meteorological forcings within the given ranges needs some more explanation. Specifically: at what time scale are the errors applied? Is it a constant offset applied to the time series for a scenario or does it have some time variability within the scenario? This should then be referred to in L. 301-306.

We explained, how uncertainties are applied to the input in more detail. We explicitly mentioned in p. 6, l.147, that a bias b was randomly chosen for each variable and then that single value applied to the variable for a given time series. Furthermore, we explicitly mentioned, that the given time series ranged from 1 October 2016 to 2 January 2017 for case WL, 3 January 2017 to 1 May 2017 for case SL and the entire season for case ALL.

Specific Comments:

At some points in the manuscript you use "snow height", but mostly "snow depth". Please use "snow depth" consistently.

For more consistency, we changed snow height to snow depth throughout the manuscript as suggested.

L. 15: "...sensitive to precipitation..."

We changed as suggested.

L. 55: You state: "However, only a few studies have so far assessed the uncertainty of snow cover models." I would rather change this to, e.g., "However, only a few studies have so far assessed the impact of forcing uncertainty on the performance of snow cover models." because there are many studies available in literature which assess the performance and uncertainty of snow cover models in general.

We changed as suggested.

L.105: "For the sensitivity analysis, we introduced uncertainties to the meteorological input." This sentence could be removed here, as you explain this in the next sections.

We removed this sentence as suggested.

L. 150: I suggest to remove the sentence "For each scenario, 14,000 simulations were performed." here, as the number of simulations is explained in the following section 2.4. You could instead extend the last sentence of 2.4 (L. 170), e.g. like "...for each of the three applied scenarios."

As suggested, we moved the content of this sentence to the next section (p.8, l.188).

We changed as suggested.

[revised manuscript text omitted]